# An engineered variant of MECR reductase reveals indispensability of long-chain acyl-ACPs for mitochondrial respiration

M. Tanvir Rahman [1], M. Kristian Koski[2], Joanna Panecka-Hofman [3,4], Werner Schmitz [5], Alexander J. Kastaniotis[1], Rebecca C. Wade [4,6], Rik K. Wierenga [1], J. Kalervo Hiltunen [1] & Kaija J. Autio [1] ✉

Mitochondrial fatty acid synthesis (mtFAS) is essential for respiratory function. MtFAS generates the octanoic acid precursor for lipoic acid synthesis, but the role of longer fatty acid products has remained unclear. The structurally well-characterized component of mtFAS, human *2E*-enoyl-ACP reductase (MECR) rescues respiratory growth and lipoylation defects of a *Saccharomyces cerevisiae Δetr1* strain lacking native mtFAS enoyl reductase. To address the role of longer products of mtFAS, we employed in silico molecular simulations to design a MECR variant with a shortened substrate binding cavity. Our in vitro and in vivo analyses indicate that the MECR G165Q variant allows synthesis of octanoyl groups but not long chain fatty acids, confirming the validity of our computational approach to engineer substrate length specificity. Furthermore, our data imply that restoring lipoylation in mtFAS deficient yeast strains is not sufficient to support respiration and that long chain acyl-ACPs generated by mtFAS are required for mitochondrial function.

Fatty acids serve a living organism as building blocks of biomembranes, energy storage, cell signaling molecules and ligands in posttranslational protein modification. Due to their economical and nutritional values there is a continuous interest in engineering of fatty acid synthesizing pathways in various host organisms to manipulate the carbon chain lengths and modifications of their products[1–3]. Frequently used approaches are heterologous expression of enzymes with different substrate specificity, generation of parallel alternative metabolic pathways or engineering of catalytic properties of endogenous enzymes of these pathways in the organisms used as biofactories. Applying structural knowledge, computer aided design and tools that allow in vivo testing of engineered variants provide avenues towards tailoring enzymes to exhibit desired properties. Aside from commercial value, appropriately designed proteins can also be helpful in the dissection of the physiological function of particular pathways.

Recently the mitochondrial fatty acid synthesis pathway (mtFAS) has been identified as a mechanism to regulate mitochondrial respiratory chain (RC) function to meet the substrate availability (acetyl-CoA) to tricarboxylic acid cycle[4,5]. MtFAS follows the prokaryotic type II mode where individual enzymatic steps are carried out by separate enzymes (Fig. 1a) Acyl groups synthesized by mtFAS are linked to 4-phosphopantetheine moiety of the acyl carrier protein (ACP) via a thioester bond. Acyl-ACPs form complexes with adapter proteins that contain a leucine-tyrosine-arginine motif (LYRM)[6]. Acyl-ACP-LYRM protein complexes facilitate assembly of mitochondrial respiratory complexes. Two acyl-ACP molecules can be found associated with LYRM proteins as components of complex I in the mitochondrial RC. Furthermore, an acyl-ACP-LYRM protein complex stabilizes the iron-sulfur cluster synthesis machinery and the LYRM protein, AltMiD51 associated with ACP, participates in the assembly of the large subunit of mitoribosomes[7–12]. MtFAS provides also the octanoic acid precursor

[1]Faculty of Biochemistry and Molecular Medicine, University of Oulu, Oulu, Finland. [2]Biocenter Oulu, University of Oulu, Oulu, Finland. [3]Faculty of Physics, University of Warsaw, Warsaw, Poland. [4]Molecular and Cellular Modeling Group, Heidelberg Institute for Theoretical Studies (HITS), Heidelberg, Germany. [5]Faculty of Biochemistry and Molecular Biology, University of Würzburg, Würzburg, Germany. [6]Zentrum für Molekulare Biologie (ZMBH), DKFZ-ZMBH Alliance and Interdisciplinary Center for Scientific Computing (IWR), Heidelberg University, Heidelberg, Germany. ✉e-mail: kaija.autio@oulu.fi

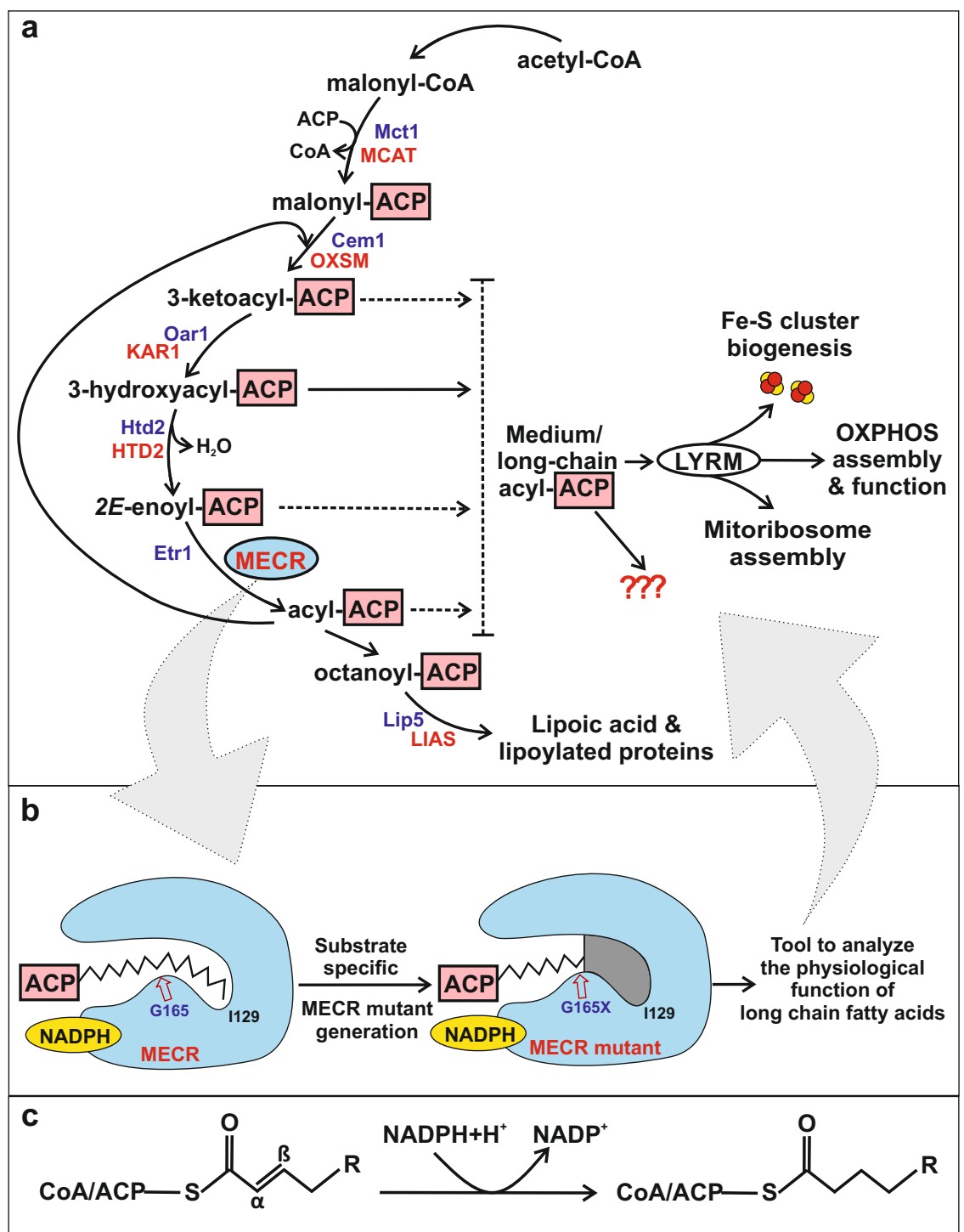

**Fig. 1 | Schematic representation of the mitochondrial fatty acid synthesis (mtFAS) pathway, wild-type and engineered MECR and the reaction catalyzed by MECR/Etr1. a** Schematic depiction of the mtFAS pathway. The indicated abbreviations (yeast (blue) /human (red)): Mct1/MCAT malonyl-CoA transferase, ACP acyl carrier protein, Cem1/OXSM 3-ketoacyl-ACP synthase, Oar1/KAR1 ketoacyl reductase, Htd2/HTD2 3-hydroxyacyl-thioester dehydratase, Etr1/MECR enoyl-thioester reductase, Lip5/LIAS lipoic acid synthetase. **b** Schematic representation of the wild-type and engineered MECR. The shown wild-type and engineered enzymes are liganded with C16- and C8-ACP molecules, respectively. The fatty acyl binding cavity extends from the catalytic site near the nicotinamide group of NADPH towards Ile129, which identifies the end of the cavity in the wild-type MECR. The engineered MECR mutant (shown as G165X) possesses a shortened substrate binding cavity discontinuing the synthesis of long-chain fatty acyl-ACP species by mtFAS. **c** MECR/Etr1 catalyzes the reduction of *2E*-enoyl substrates to their saturated counterparts in a NADPH-dependent manner. MECR accepts fatty acyl groups that are attached to either CoA or ACP via a thioester bond.

required for the endogenous synthesis of lipoic acid, an essential cofactor in oxidative decarboxylation of α-ketoacids and glycine[13]. MtFAS is capable of generating fatty acids longer than eight carbons, but the precise function of these longer fatty acids has remained unclear. We are interested in studying the function of these long-chain acyl groups synthesized by mtFAS and associated with ACPs. The most convenient model to investigate basic aspects of mtFAS function is yeast *Saccharomyces cerevisiae*, serving as a fast read-out system and

the function of a deleted yeast mtFAS enzyme can be replaced by counterparts from heterologous sources[14,15].

The last step of mtFAS is carried out by *2E*-enoyl-ACP thioester reductases MECR and Etr1 in human and yeast, respectively. These enzymes are members of the medium-chain dehydrogenase / reductase (MDR) superfamily[16], catalyzing the NADPH-dependent reduction of *2E*-enoyl thioesters into acyl thioesters (Fig. 1c)[17,18]. Wild-type human MECR complements the yeast respiratory deficient phenotype of the *Δetr1* strain. The crystal structures of the unliganded enzymes of human MECR and the *Candida tropicalis* Etr1, as well as the structures of the NADPH binary complex, and the NADPH-crotonoyl-CoA ternary complex of the latter enzyme have been solved[19–21]. The structure of MECR shows a bent cavity of sufficient dimensions to accommodate acyl substrates with carbon chain lengths from C4 to C16[19]. In vitro,

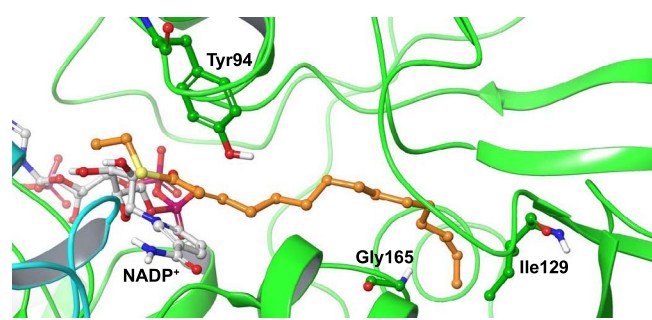

**Fig. 2 | The model of the wild-type MECR.** The wild-type MECR holoenzyme was modeled based on the crystal structure with PDB entry 2VCY; the cofactor NADP+ (white carbons) and the C16-substrate fragment (orange carbons) were modeled-in based on the *C. tropicalis* ETR1 crystal structure (PDB entry 4WAS). The two residues mutated in this work and the catalytic residue Tyr94 are shown. The two subunits of the MECR enzyme dimer are displayed as cartoon colored green and cyan. Non-polar hydrogens are not shown.

MECR accepts fatty acyl groups that are attached to either CoA or ACP via a thioester bond as substrates.

Here, we describe our work on molecular modeling and simulations guiding the engineering of MECR in order to obtain variants that are unable to accept long-chain fatty acyl substrates. We tested these variants in vivo in Etr1-deficient yeast followed by in vitro studies (Fig. 1b). Our results show that unlike the wild-type human MECR, the engineered G165Q variant of MECR did not rescue the yeast respiratory deficient phenotype of the *Δetr1* strain, although protein lipoylation was restored, demonstrating that the octanoyl/lipoyl synthesizing branch of the mtFAS is supported by enzyme variant. Our data indicate that provision of the octanoic acid precursor by mtFAS is not sufficient to support mitochondrial function and long acyl tail(s) must be generated by mtFAS to allow the cells to maintain a respiratory competent mitochondrial population.

## Results

To study the physiological role of long-chain fatty acids generated by mtFAS, we generated a yeast strain that cannot synthesize long-chain fatty acids in mitochondria. This was done by engineering the fatty acyl binding cavity of human MECR based on in silico modeling and further in vivo and in vitro studies of the MECR variants.

### Computational design of the MECR mutants

Computational modeling was initiated by building the MECR/NADP+ complexes with a series of substrates from C8 (octanoic acid) to C16 (Fig. 2 and Supplementary Fig. 1). Potential substitution points, where mutations will block the long chain substrate binding (while not distorting the binding of *2E*-octenoyl-CoA and shorter substrates) were evaluated (Supplementary Tables 1 and 2). Therefore, the following approaches were considered: (i) substitution of amino acid residues in the cavity by bulkier hydrophobic residues that block the space for the hydrophobic alkyl substrate chains, and (ii) modification of the above amino acid residues to polar/charged moieties to create a stable interaction network shortening the cavity. We have previously generated variants I129M, F324Y and G165S in MECR and in vitro analysis of their kinetic properties showed that these variants were capable of

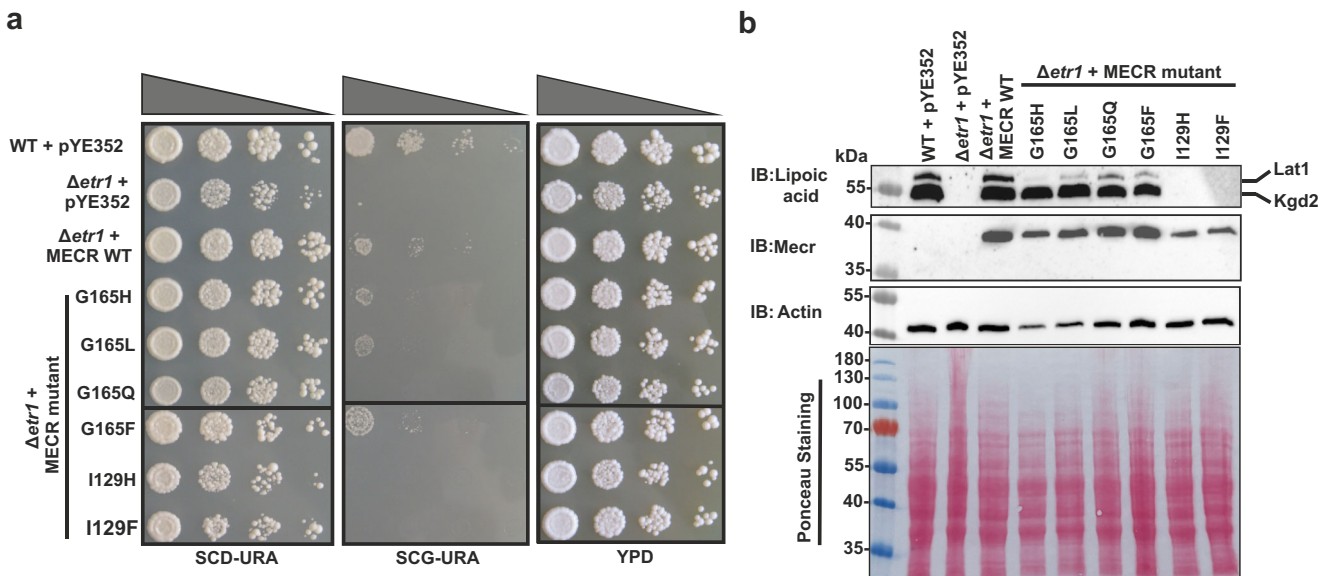

**Fig. 3 | In vivo screening of human MECR mutants in yeast. a** Ten-fold dilution series testing for growth of wild-type and mutated human MECR variants on glucose (SCD-URA and YPD) and glycerol (SCG-URA) growth media. First row: Wild-type Bj1991α yeast strain with empty vector pYE352, Second row: Null mutant (*Δetr1*) Bj1991α yeast strain with empty vector pYE352, third row: Bj1991α *Δetr1* yeast strain with mitochondrially targeted MECR in pYE352, fourth row to nineth

row: Bj1991α *Δetr1* with MECR single mutant in pYE352. **b** Western blot analysis of protein lipoylation and MECR expression in BJ1991α yeast expressing wild-type and mutated MECR variants. Actin and Ponceau staining were used as loading controls and the experiments were repeated three times with individual biological samples. Source data are provided as a Source data file.

reduction of the long chain *2E*-enoyl-CoA substrates[19]. Visual inspection of the whole active site identified I129 and G165 as the most promising substitution sites (Fig. 2) that could discriminate between *2E*-C8 and longer substrates, while F324 was considered to be located too far from the predicted location of C8 of the *2E*-enoyl tails. I129 and G165 are conserved in the MECR sequences of higher eukaryotes (Supplementary Fig. 2). I129 is positioned in the bottom of the fatty acyl binding cavity at the end of β-strand β5, whereas G165 is located in the α-helical region in the center of the cavity between helices αD and αa (Figs. 1b, 2). The backbone geometry of the modeled MECR at position 165 (Phi/Psi angles = −70°/−18°) would allow for substitution by a residue with a side chain, as would the backbone geometry at position 129 (−122°, 159°). Accordingly, I129 and G165 were computationally replaced with amino acid residues satisfying the aforementioned design assumptions. This approach allowed us to prioritize the following MECR single-point mutations: G165H, G165L, G165Q, G165F, I129H and I129F (Supplementary Fig. 3). In silico models showed more steric clashes for G165H and G165F than for the other G165 mutants, but these interferences occurred with residues located in the loops. All these six mutants were chosen for in vivo screening.

## In vivo screening of the MECR mutants in yeast

The mutations proposed by in silico modeling were cloned into a pYE352 yeast expression vector containing human *MECR* cDNA and N-terminally appended with a yeast mitochondrial targeting sequence[15]. These constructs were introduced into the Δ*etr1* yeast strain, and growth properties as well as protein lipoylation were examined to determine the effects of the amino acid substitutions in an in vivo system. It has been previously established that respiratory deficient phenotype of Δ*etr1* strain can be complemented by mitochondrially targeted human MECR[15]. Growth of the transformants was tested on fermentable (glucose) and non-fermentable (glycerol) carbon sources (Fig. 3a). All strains grew wild-type or near wild-type levels on synthetic complete dextrose (SCD) and YPD media, which are fermentable media. Yeast strains carrying MECR with G165H, G165L or G165F mutation were viable on non-fermentable, glycerol medium (SCG), while mutants G165Q, I129H and I129F did not grow or grow very poorly on non-fermentable, glycerol medium, indicating respiratory deficient phenotype.

Because mtFAS produces octanoic acid for lipoid acid synthesis, the ability of the mutants to generate mitochondrial fatty acids up to C8 was examined by western blot using anti-lipoic acid antibody (Fig. 3b). The results show that mutations of G165 allow protein lipoylation of Kgd1 but not of Lat1, indicating that MECR mutants G165H, G165L, G165Q and G165F are able to synthesize fatty acids at least up to C8. Yeast strains carrying MECR mutants I129H or I129F did not show any lipoylated proteins. Nevertheless, in all yeast strains expressing wild-type MECR or its point mutated variants, western blot analysis confirmed the presence of the expressed MECR protein (Fig. 3b). Taken together, this in vivo analysis demonstrates that the mutation G165Q in MECR allows Δ*etr1* strain to synthesize lipoic acid but does not restore respiratory competence.

## Protein purification and analysis

Based on in silico prediction and in vivo screening the MECR G165Q variant was chosen for further functional and structural studies. Wild-type and mutant G165Q MECR appended with a C-terminal His-tag were expressed in *Escherichia coli* and purified to apparent homogeneity with Ni-NTA affinity chromatography, followed by size-exclusion chromatography. The circular dichroism (CD) spectra show that the gross secondary structure elements of wild-type MECR and MECR G165Q proteins are congruent (Supplementary Fig. 4a). Thermal stability titration curves are similar for both proteins and the calculated melting points are $T_m$ = 45.4 °C and $T_m$ = 45.7 °C for wild-type MECR and MECR G165Q, respectively (Supplementary Fig. 4b).

**Table 1 | Kinetic parameters of wild-type MECR and the MECR G165Q variant**

| Substrate | Kinetic parameters | Wild-type MECR | MECR G165Q variant |
|---|---|---|---|
| 2*E*-hex-enoyl-CoA | $K_M$ (µM) | 90.6 ± 26.5 | 111.2 ± 17.3 |
| | $k_{cat}$ (s⁻¹) | 0.99 ± 0.22 | 3.49 ± 0.33 |
| | $k_{cat}/K_M$ (M⁻¹s⁻¹) | 0.01 × 10⁶ | 0.03 × 10⁶ |
| 2*E*-octe-noyl-CoA | $K_M$ (µM) | 23.2 ± 5.4 | 114.1 ± 16.2 |
| | $k_{cat}$ (s⁻¹) | 1.80 ± 0.13 | 12.45 ± 1.90 |
| | $k_{cat}/K_M$ (M⁻¹s⁻¹) | 0.08 × 10⁶ | 0.11 × 10⁶ |
| 2*E*-dece-noyl-CoA | $K_M$ (µM) | 15.9 ± 2.2 | 8.15 ± 1.54 |
| | $k_{cat}$ (s⁻¹) | 0.62 ± 0.02 | 0.30 ± 0.02 |
| | $k_{cat}/K_M$ (M⁻¹s⁻¹) | 0.04 × 10⁶ | 0.04 × 10⁶ |

The assays were done as triplicate, and values are given as mean ± standard deviation. Source data are provided as a Source data file.

These results indicate that the G165Q mutation does not affect the overall structure and stability of MECR.

## Analysis of kinetic properties of wild-type and G165Q MECR

To investigate how the G165Q mutation affects the catalytic properties of MECR, we determined the kinetic parameters of wild-type and G165Q variants of MECR (Table 1). *2E*-enoyl-CoA thioesters with varying fatty acyl tail lengths (C6, C8 and C10) were used as substrates. Because long chain *2E*-enoyl-CoA esters are poor substrates for the G165Q variant, only specific activities were determined for *2E*-dodecenoyl-CoA, *2E*-tetradecenoyl-CoA and *2E*-hexadecenoyl-CoA substrates (Table 2). Interestingly, the $k_{cat}$ of G165Q MECR with *2E*-octenoyl-CoA substrate was 6.9 times higher than the $k_{cat}$ of wild-type protein. The $K_M$ values of wild-type MECR and the G165Q mutant enzyme decrease from 90.6 to 15.9 µM for wild-type and 111.2 to 8.15 µM respectively when the chain length of the substrate increases from six carbons (C6) to ten carbons (C10) (Table 1). Thus, the $K_m$ value of wild-type MECR with C10 substrate is 18% of the $K_m$ value with C6 substrate, while in G165Q mutant enzyme the $K_M$ value with C10 substrate is 7% of the $K_M$ value with C6 substrate. The systematic variation of the $k_{cat}$ and $K_M$ values for the C6, C8 and C10 substrates is intriguing, but it is difficult to correlate these variations with the structural properties of wild-type MECR and its G165Q variant. Due to the complicated catalytic cycle of the MECR reaction the physical meaning of $k_{cat}$ is not known (it could be related to the off dissociation of the product), likewise the $K_M$ depends on the relative rates of the different steps of the reaction and its value cannot be directly correlated with the affinity of the substrate.

Both wild-type and the G165Q variant were able to accept C10, C12 and C14 substrates, but the low catalytic rates for the G165Q variant did not allow characterization of kinetic properties in detail for the C12 and C14 substrates. Of note, the observed catalytic rate for the G165Q variant was eight times lower with C12 substrate and six times lower with C14 substrate when compared to the wild-type enzyme. The activity of the G165Q variant with C16 substrate was below the detection limit. The residual activity with the C12- and C14-tail substrates could well be related to the increased structural flexibility of the G165Q variant, as suggested by the B-factor properties (Supplementary Fig. 6).

## Crystallographic studies of MECR

Next, we aimed to obtain information on structural features resulting in diminished catalytic efficiency with long chain enoyl fatty acid substrates due to the G165Q point mutation in MECR. G165Q variant

**Table 2 | Specific activities of wild-type MECR and the MECR G165Q variant with C12, C14 and C16-CoA substrates**

| Substrate | Substrate concentration (µM) | Turnover numbers of wild-type MECR (s⁻¹) | Turnover numbers of the MECR G165Q variant (s⁻¹) |
|---|---|---|---|
| 2E-dodecenoyl-CoA | 30 | 0.75 ± 0.08 | 0.09 ± 0.01 |
| 2E-tetradecenoyl-CoA | 30 | 0.35 ± 0.03 | 0.06 ± 0.01 |
| 2E-hexadecenoyl-CoA | 30 | 0.28 ± 0.06 | Below detection level |

The activities of MECR for long-chain enoyl substrates are too low for accurate determination of the Michaelis-Menten kinetic parameters. Therefore, the specific activities (expressed as turnover numbers) for fixed substrate concentration are given. The assays were done as triplicate, and values are given as mean ± standard deviation. Source data are provided as a Source data file.

did not provide crystals under conditions previously used in MECR crystallization[19], so completely new crystallization parameters were screened. Both wild-type and the G165Q MECR formed hexagonal crystals at +4 °C in a condition where the main precipitation agents were PEG 6000 and NaCl and the pH was 4.5 as described in Material and Methods. Crystal structures were solved at 1.85 Å and 2.02 Å resolution, respectively, in the trigonal space group $P3_121$ (Fig. 4, Supplementary Table 3). The asymmetric unit contains one MECR chain, but the enzyme forms a dimer via a crystallographic 2-fold axis. Compared to the available structures of human MECR (PDB entry 2VCY, 2.41 Å or PDB entry 1ZSY, 1.75 Å), the condition, space group and the crystal packing are different. Previously, human MECR crystals were obtained at room temperature, at pH 8.5 (2VCY) or pH 6.5 (1ZSY) under ammonium sulfate-based conditions, and the crystals exhibited tetragonal (2VCY, space group $P4_22_12$) or orthorhombic (1ZSY, space group I222) symmetry lattices. The 2VCY structure includes a MECR dimer in the asymmetric unit[19], whereas 1ZSY has only one subunit in the asymmetric unit. The dimeric structure of the MECR trigonal crystal form of this study, obtained by using the crystal symmetry, is similar to the dimeric structure (found in the asymmetric unit) of the tetragonal MECR crystal form.

The overall fold of the MECR structures determined in this study is maintained, with RMSD value of 0.63 Å for the Cα-atoms of 332 aligned residues (Fig. 4), indicating that the structure of MECR G165Q is not changed. This conclusion is also supported by CD analysis data (Supplementary Fig. 4a). In the trigonal crystal form, MECR adopts the closed conformation, previously observed in MECR structures 2VCY and 1ZSY, as well as in the CtEtr1p structure complexed with NADPH[20]. In the MECR structures presented here, the first N-terminal residues (residues A31-A41) are not defined by the electron density maps, but the C-terminus (till L374) including the six residue long His-tag sequence is seen well. In the tetragonal and orthorhombic crystal forms determined earlier, the last visible residue is M373. The first visible residue in the current and 2VCY structures is R42. Otherwise, the current and previous wild-type structures are very similar (RMSD 0.70 Å for 332 aligned residues). There are some small differences in certain loop regions such as 240-252. In the current structure this region has high B factors, but residues are still well defined by the electron density maps. This region had very weak and fragmented structure in the previous tetragonal MECR structure. This loop has been proposed to be important for the recognition of the ACP moiety of the substrate molecule.

The electron density map of the MECR G165Q crystal structure shows the structural changes due to the mutation (Fig. 4b). The structure of the Q165 side chain is well defined by the electron density map (Supplementary Fig. 5). The mutation site locates in the domain interface in between the α helix αD of the catalytic domain and α helix αA of the cofactor binding domain. The side chain points across the acyl tail binding tunnel, making a good hydrogen bond between NE2(Q165) and O(P130). This interaction causes changes in the main chain conformation of the residues near P130 and Q165, as well as of the side chain conformations of K316 and E107. In wild-type MECR, the side chain of E107 is hydrogen bonded to backbone nitrogen atoms of G165 and V166, whereas in the MECR G165Q, the side chain of E107 is hydrogen bonded to backbone nitrogen of V166. Furthermore, in wild-type MECR, the side chain of K316 is hydrogen bonded to the

backbone oxygen of G165 (Fig. 4d), but in the MECR G165Q variant, the lysine side chain is hydrogen bonded with the side chain of Q165 and the backbone oxygen of P130 (Fig. 4e). The hydrogen-bonding network around Q165 is somewhat different in the experimental crystal structure as compared to the computational model of MECR G165Q (Supplementary Fig. 7). The MECR G165Q structure has a higher overall B factor than wild-type MECR (Supplementary Table 3). Especially the region close to the catalytic Y94 moiety, which has interactions with the loop regions near P130 and E107, has higher B factors compared to the corresponding wild-type structure (Supplementary Fig. 6).

## Molecular docking of substrates to wild-type and G165Q MECR variants

Finally, we analyzed the effect of G165Q mutation on substrate binding by in silico docking studies, presenting 2E-enoyl substrate fragments with the length of C8-C16 (Supplementary Fig. 1) to the wild-type MECR (Supplementary Fig. 8) and MECR G165Q (Supplementary Fig. 9) structures, both crystallized in this work. The aligned docking results of substrate fragments to the wild-type MECR and G165Q variant are presented in Supplementary Fig. 10. These dockings were made to dimeric MECR structures where the B chain was generated using the crystallographic symmetry operators. The results obtained from these modeling studies are summarized in Supplementary Table 4. In the wild-type structure, substrates up to C16 are positioned in the previously identified substrate tunnel (Supplementary Fig. 8), which is overall consistent with the data from studies of MECR[19]. In the MECR G165Q variant pocket, for the 2E-C14 substrate no docking poses were returned and the poses of 2E-C12 and 2E-C16 did not reach the end of the binding pocket (Supplementary Fig. 9d, e), due to the blockage created by the side chain of Q165. Clearly, the G165Q mutation blocks the acyl tail binding tunnel (Fig. 4e), in good agreement with the changed substrate specificity.

The hydrocarbon chains of the docked substrates have multiple unrestricted rotational degrees of freedom. We assumed that the docking solutions obtained represent the ensemble of accessible conformations of the flexible substrate hydrocarbon chains. In wild-type, for 2E-C8 to 2E-C14, the number of poses is similar (on average 8, Supplementary Table 4), whereas for the longest substrate, 2E-C16, just one pose was obtained. In contrast, the number of substrate docking poses for the MECR G165Q variant decreases with the increasing substrate length, starting from 2E-C8. This confirms that the more confined pocket of the G165Q variant reduces the number of possible states of longer substrate chains and suggests an unfavorable conformational entropy contribution to longer substrate binding in the MECR G165Q variant.

The average RMSD values of the docked poses with respect to the substrate core fragment and selected energy estimates over each ensemble of the docked poses were analyzed for each ligand and for each docking variant (Supplementary Table 4). The average docking scores (estimates of substrate binding free energies, negative is favorable) are only slightly negative, with the exception of the unfavorable positive values for the poses of 2E-C12 and 2E-C16 substrates in the G165Q variant. These relatively high docking scores are probably reflective of the importance of receptor conformational changes upon substrate binding, which are not accounted for in the docking simulations. The average scores were less favorable in the MECR G165Q variant

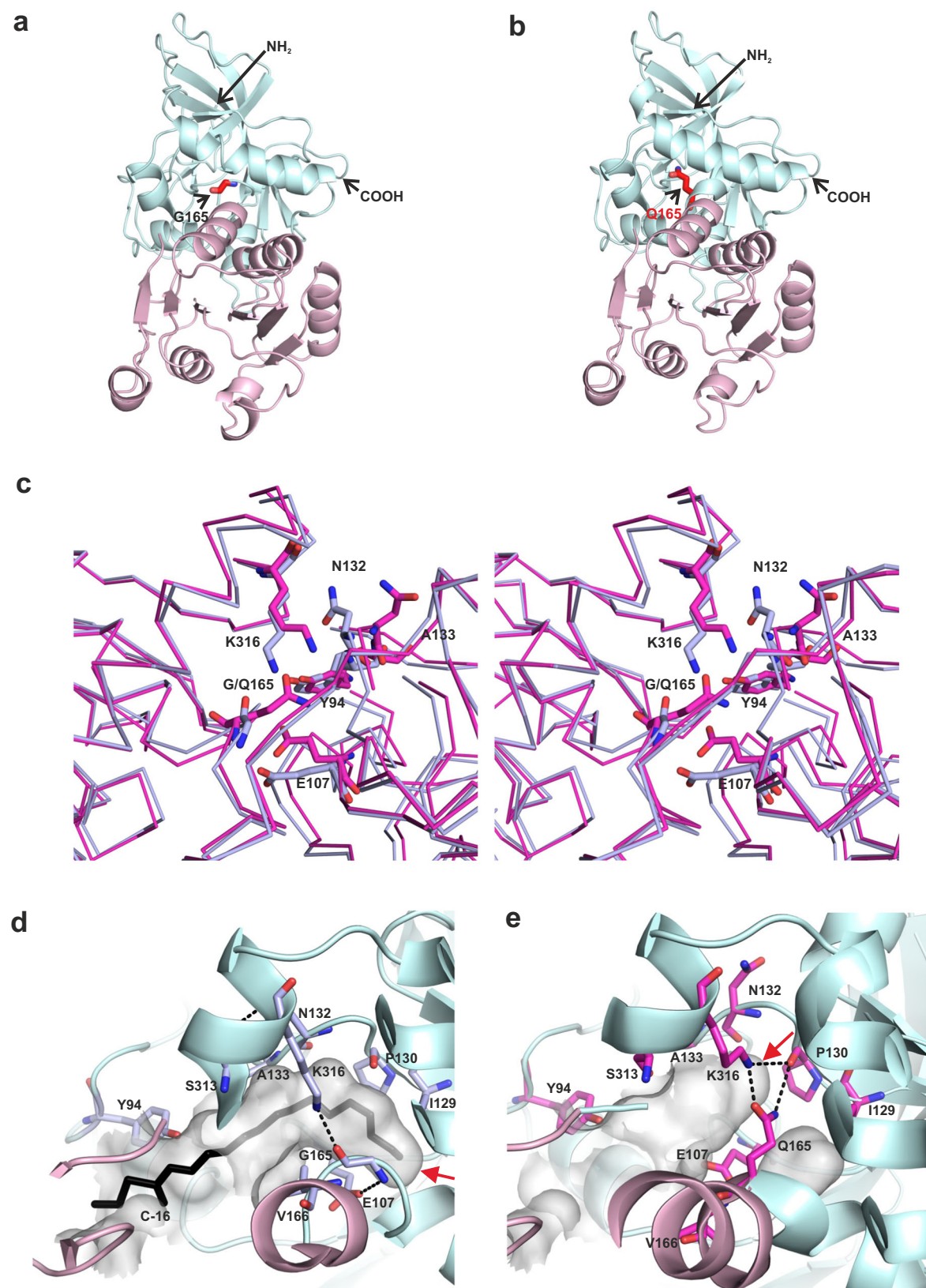

than wild-type for all the docked substrates that returned any poses. However, it is worth noting that 2*E*-C8 and 2*E*-C10 have slightly more favorable average van der Waals energy contributions to binding in the G165Q variant than in wild-type (2*E*-C8: −27.1 vs. −26.7 kcal/mol, 2*E*-C10: −32.0 vs. −28.9 kcal/mol, respectively), in contrast to 2*E*-C12 (−25.0 vs. −31.4 kcal/mol, respectively) and 2*E*-C16 (−17.0 vs. −31.3 kcal/mol,

respectively). The ligand average internal energy for 2*E*-C8 and 2*E*-C10 is also distinctly more favorable (lower) in the G165Q variant than in wild-type (2*E*-C8: 2.8 vs. 4.5 kcal/mol and 2*E*-C10: 3.4 vs. 5.9 kcal/mol, respectively). This contrasts with 2*E*-C12 and 2*E*-C16, for which this relation is inverted (2*E*-C12: 8.7 vs. 6.5 kcal/mol and 2E-C16: 13.2 vs. 6.2 kcal/mol, respectively).

**Fig. 4 | The structure of human wild-type MECR and the MECR G165Q variant.** Crystal structure of the (**a**) wild-type MECR of this study (PDB entry 7AYB), and (**b**) the MECR G165Q mutant (PDB entry 7AYC). Only one monomer of the dimeric MECR is shown. The catalytic domain is colored pale cyan and the cofactor binding domain is colored light pink. The mutation site is highlighted. NH2 and COOH label the N-terminus and C-terminus. **c** The structural comparison of the superposed wild-type MECR and MECR G165Q variant. The stereo diagram showing the conformational changes in MECR G165Q variant structure due to the mutation. Residues of wild-type protein are shown in light blue color and the residues of mutant protein are labeled in magenta color. The side chain of the selected residues are shown in stick representation with N and O atoms colored blue and red, respectively. It can be noted that the (phi/psi) values of G165 in wild-type would allow for the mutation into a residue with a side chain: (phi/psi) G165 (wild-type) = −73/−11; (phi/psi) Q165 (G165Q) = −84/−55. **d** Detailed view of the fatty acyl binding cavity of wild-type MECR with a modeled C16-substrate. The cavity for fatty acyl tail is shown as "cavity surface" option (light gray + transparent) and the bottom of the cavity is shown with red arrow. **e** Fatty acyl binding cavity in the G165Q structure is shown as "cavity surface" option (light gray + transparent) and the bottom of the cavity is shown with red arrow. Hydrogen-bonding network is changed significantly and the loop region containing A133 and N132 moves slightly as also seen in the panel c.

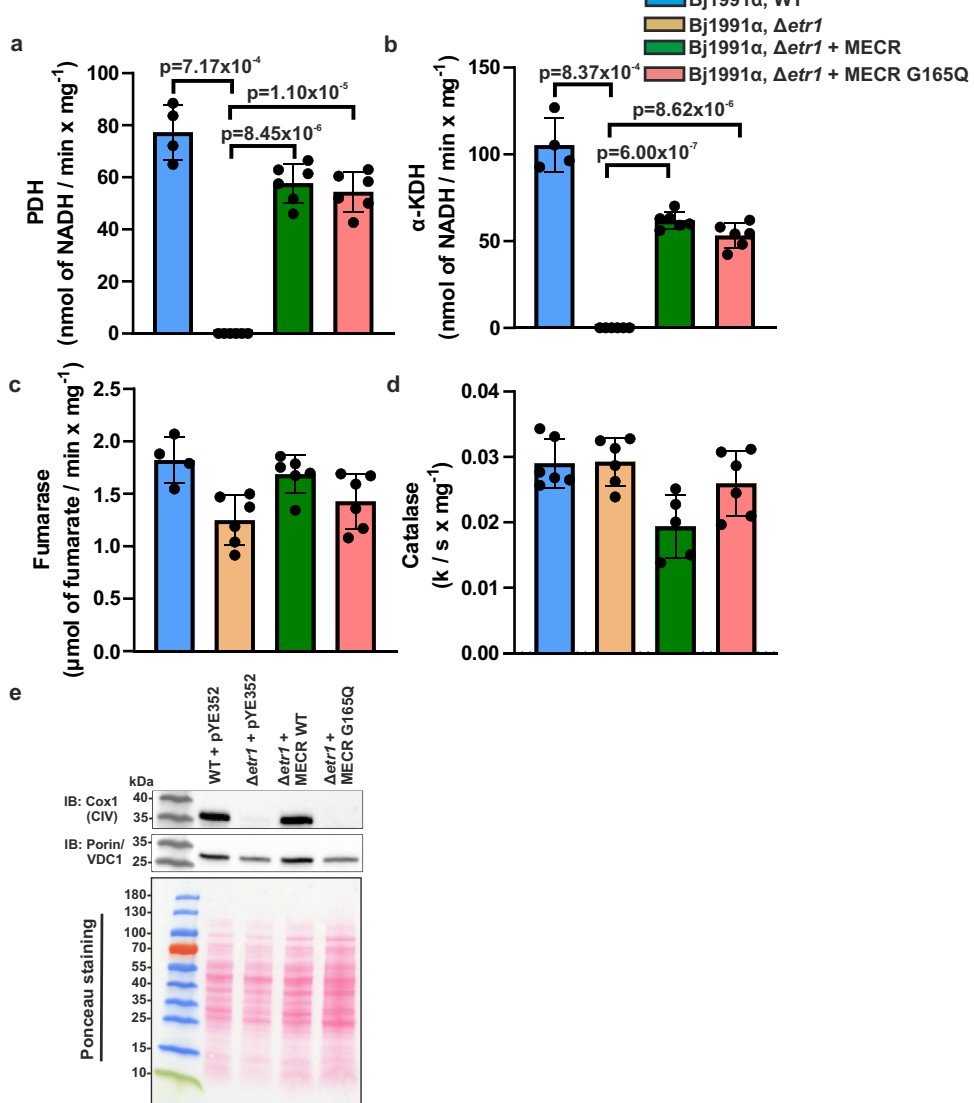

**Fig. 5 | Pyruvate dehydrogenase (PDH), α-ketoglutarate dehydrogenase (α-KDH), fumarase and catalase activities and expression of Cox1.** Enzyme activities for (**a**) PDH, (**b**) α-KDH and (**c**) fumarase were analyzed from wild-type (WT), *Δetr1* and *Δetr1* expressing wild-type MECR or the G165Q variant mitochondria. **d** Catalase activity from wild-type (WT), *Δetr1* and *Δetr1* expressing wild-type MECR or the G165Q variant was analyzed from yeast spheroplasts. All experiments were performed in duplicate. The number of biological repeats in PDH and KDH activity assays is four for wild-type yeast mitochondria, five for *Δetr1* strain, and six for *Δetr1* expressing wild-type MECR or G165Q variant. The number of biological repeats in fumarase activity assays is four for wild-type yeast mitochondria, six for *Δetr1* strain, and six for *Δetr1* expressing wild-type MECR or G165Q variant. The number of biological repeats in catalase activity assays is six for wild-type yeast mitochondria and *Δetr1* strain, five for *Δetr1* expressing wild-type MECR, and six for *Δetr1* expressing MECR G165Q variant. Data were analyzed by two-tailed, unpaired Student's *t*-test and the results are expressed as mean ± standard deviation (SD). The enzyme activity data were plotted by using GraphPad Prism computer software. **e** Steady state level of Cox1 in isolated yeast mitochondria was analyzed by western blotting. Porin was used as a loading control. Source data are provided as a Source data file.

The average RMSD values (Supplementary Table 4) for the substrate core for 2*E*-C8 and 2*E*-C10 are lower in the G165Q variant than in wild-type (2*E*-C8: 0.5 vs 0.7 Å, 2*E*-C10: 0.5 vs 0.8 Å, respectively), while higher for 2*E*-C12 and 2*E*-C16 (2*E*-C12: 0.9 vs. 0.6 Å, 2*E*-C16: 0.8 vs. 0.7 Å). This implies that the catalyzed moiety of the substrate is more "correctly" positioned for 2*E*-C8 and 2*E*-C10 in the G165Q variant than in wild-type.

Taken together, the docked 2*E*-C8 and 2*E*-C10 ligands are less strained in the G165Q variant than in wild-type (internal energy), and both 2*E*-C8 and 2*E*-C10 have slightly better steric fit (van der Waals energy) and lower RMSD of the catalytic core in the G165Q active site than in wild-type. The above observations are consistent with the slightly lower $K_M$ of 2*E*-C10 in the G165Q variant as compared to wild-type and the higher $k_{cat}$ for 2*E*-C8 in G165Q than in wild-type (Table 1).

### Effect of MECR G165Q mutation on the total fatty acid profile and Cox1 expression

The effect of MECR G165Q mutation on cellular fatty acids was studied by liquid chromatography / mass spectrometry. There were no changes in cellular fatty acid profiles between wild-type, Δ*etr1* or Δ*etr1* cells carrying plasmid expressing MECR or the G165Q variant (Supplementary Fig. 11). The majority of cellular fatty acids are produced by cytosolic FAS I pathway and the contribution of mtFAS to the total cellular FA pool is negligible[22], thus this result was expected. To confirm that octanoyl-ACP and lipoic acid are synthetized in our yeast strains, we measured pyruvate dehydrogenase (PDH) and α-ketoglutarate dehydrogenase (α-KDH) activities from mitochondrial extracts (Fig. 5a and b). PDH activities were $77 \pm 11$, $58 \pm 8$ and $54 \pm 8$ (mean $\pm$ SD) nmol of NADH / min x mg$^{-1}$ for wild-type yeast mitochondria, for the Δ*etr1* strain expressing MECR and for the Δ*etr1* strain expressing MECR G165Q variant, respectively. Correspondingly, α-KDH activities were $105 \pm 15$, $62 \pm 5$ and $53 \pm 7$ (mean $\pm$ SD) nmol of NADH / min x mg$^{-1}$ for wild-type, for the Δ*etr1* strain expressing MECR and for the Δ*etr1* strain expressing MECR G165Q variant. Both PDH and α-KDH activities were below detection limit in Δ*etr1* strain. In both assays the number of biological replicates was 4 for wild-type yeast mitochondria, 5 for the Δ*etr1* strain, 6 for the Δ*etr1* strain expressing MECR and 6 for the Δ*etr1* strain expressing MECR G165Q variant. Fumarase activity was measured from all samples to ensure the quality of mitochondrial extracts and no changes between the samples were detected (Fig. 5c). α-KDH generates succinyl-CoA that is needed in heme synthesis. In previous studies, heme deficiency due to the non-functional α-KDH caused by the lack of lipoic acid was discussed as a possible cause of the respiratory-defect phenotype in ΔmtFAS yeast strains[4]. Catalase is a heme containing enzyme. To assess the heme content in yeast cells, we analyzed catalase activities in spheroplasts (Fig. 5d). We were not able to detect any significant changes in catalase activity in wild-type, Δ*etr1* and Δ*etr1* expressing wild-type MECR or G165Q variant.

It has been shown previously that defects in mtFAS like deletion of *Etr1* in yeast lead to severely reduced levels of Cox1, a mitochondrially encoded subunit of respiratory complex IV[4]. It was also concluded that reduced Cox1 is due to a decrease in Cox1 translation caused by a respiratory complex IV assembly defect. Here we employed the investigation of Cox1 abundance as a readout to obtain information on the cause of the respiratory deficiency of Δ*etr1* yeast strain expressing the G165Q MECR mutant. Extracts of the Δ*etr1* strain transformed with the plasmid carrying the *MECR* G165Q mutation were analyzed by western blot using Cox1 antibody (Fig. 5e). In agreement with the previous results, Cox1 was visualized in the extract from wild-type yeast cells, but barely detectable from Δ*etr1* cell extract. Cox1 was visualized on the wild-type level on extract from Δ*etr1* cells expressing wild-type MECR, but the extract from Δ*etr1* cells expressing G165Q mutant MECR was depleted of Cox1. Because expression of Cox1 in Δ*etr1* and Δ*etr1* cells expressing G165Q variant was negligible, one possibility is that the cells have lost their mitochondrial DNA. This was

tested by introducing back *Etr1* in a plasmid and then following the respiratory growth on glycerol plate (Supplementary Fig. 12). Previously it has been shown that Δ*etr1* cells after introduction of Etr1 back are able to grow on SCG-plates[14]. Here we show that also Δ*etr1* strain expressing MECR G165Q variant is able to grow on SCG-plate after introduction of Etr1. Thus, these data suggest that both the strains are rho$^+$.

## Discussion

MtFAS acts as an integrator between mitochondrial availability of acetyl-CoA and respiratory chain function[4,5]. To mediate this coordinator function, acyl-ACPs generated by mtFAS interact with LYRM adapter proteins and these complexes facilitate assembly and functions of several components of the respiratory chain, mitoribosomes and in Fe-S cluster biogenesis. In addition to these LYRM protein dependent actions, mtFAS provides the octanoyl-groups that are used in endogenous lipoic acid synthesis and is essential for lipoylation and function of mitochondrial α-keto acid decarboxylase complexes. Due to this multiplicity in function, deficiencies in mtFAS in humans result in a pleiotropic phenotype of affected individuals. Of note, currently there is no evidence that mtFAS provides fatty acyl groups to structural lipids, with the exception of a lipid A-like molecule described in plants[23].

A study published by Brody and Mikolajczyk[24] showed that mitochondrial ACP isolated from the fungus *Neurospora crassa* carries the 3-hydroxytetradecanoyl-group. Proteolytic analysis of respiratory complex I in bovine heart mitochondria by Carroll et al. in 2003[25] revealed that ACP in complex I contained an extra mass that suits 3-hydroxytetradocanoic acid. More recently, cryo-electron microscopy structural studies of the porcine heart complex I identified a decanoyl-phosphopantetheine group on ACPs in complex with LYRM proteins NDUFA6/LYRM6 or NDUFB9/LYRM3[26]. All these data indicate that ACP in complex I can be attached to a long-chain acyl group. Additionally, ACP has been reported to interact with the LYRM protein ISD11/LYRM4 in iron-sulfur cluster biogenesis. Both structural and mass spectrometric analysis carried out by multiple groups have shown that this ACP-ISD11 interaction is supported by a long-chain fatty acid bound to ACP[8,27–29]. It has remained unclear if short chain acyl groups attached to ACP are sufficient to mediate ACP-LYRM protein functions or if acyl groups longer than C8 are indispensable for respiration.

To shed light on the role of long-chain fatty acids produced by mtFAS, we generated a yeast strain in which synthesis of mitochondrial long-chain fatty acids was abolished by mutating the substrate binding cavity of enoyl reductase catalyzing the last step of mtFAS. In silico molecular modeling methods were used to identify mutations with the potential to exhibit the desired properties. Six mutations suggested by the models to shorten the substrate binding cavity were tested in vivo for complementation of the respiratory and lipoylation deficient phenotypes of the Δ*etr1* yeast strain. The MECR G165Q mutation was chosen for further analysis on the following grounds: (i) The immunoblotting experiments indicated that the mutated human MECR variant was expressed in the Δ*etr1* yeast strain. (ii) The yeast strain expressing the MECR G165Q variant showed protein lipoylation suggesting that mtFAS was functional and fatty acids to minimum length of eight carbons were synthesized. (iii) However, the testing growth on the glycerol as carbon source, showed the strain was not respiratory competent. At this point we concluded that the respiratory deficiency of the Δ*etr1* yeast expressing the MECR G165Q variant was due to changes in stability, structure or kinetic properties of the engineered protein. To differentiate these options, the MECR G165Q was further characterized in vitro. We expressed and purified both wild-type and MECR G165Q and analyzed their CD spectra and thermal stability. These studies showed that G165Q mutation in MECR does not affect the structure and the stability of the protein. Enzyme kinetic

studies indicated that the G165Q mutation resulted in increase in catalytic activity when C8 was used as a substrate, but there was no effect with the C10 substrate and the activities were significantly lower with C12 and C14 substrates. The changes in enzymatic activity of MECR G165Q variant are similar to the previously analysed G165S mutant: both mutations increase the activity when using C8 substrate but the activities with longer fatty acyl substrate like C14 are much lower when compared to wild-type[19].

We also solved the crystal structures of wild-type and G165Q mutant MECR at higher resolution compared to earlier studies[19]. This structure agreed quite well with the modeling studies of the G165Q variant, though there are some differences in the hydrogen-bonding network that involve Q165, and some other residues in the catalytic domain, which were not predicted by modeling (Supplementary Fig. 7). In the wild-type MECR, the substrate binding cavity can accommodate fatty acyl substrate longer than C8 (Fig. 4d). The introduction of the glutamine side chain to the position 165 neither affected the overall structure, nor the conformation of the catalytic tyrosine (Y94) or the NADPH binding site, but together with conformational changes of an acidic (E107) and basic (K316) amino acid, the modification induces a change of the acyl binding cavity of the mutated protein (Fig. 4e). As a result, the fatty acyl binding cavity is much shorter in the MECR G165Q variant (Fig. 4e). We also analyzed the binding of various chain-length substrates to both wild-type and G165Q mutant structures by in silico docking studies. Overall, the average docking score values suggested less favorable binding of substrates to the G165Q mutant. Notably, the C8 substrate exhibited a more favorable average internal energy and van der Waals energy in the G165Q mutant than in the wild-type MECR. Also, the catalyzed substrate "core" was closer to the crystallographic position for C8 in the mutant than in wild-type. We hypothesize that the latter observations might be related to the fact that the C8 substrate is catalyzed faster in the G165Q mutant than in wild-type (Table 1).

Only the G165Q variant had the desired properties, whereas the variants with the G165H, G165L, G165F point mutations could still process long-chain substrates. Most likely, this is related to the unique hydrogen bond properties of the Q165 side chain (Supplementary Fig. 7), providing the acyl tail binding tunnel with its desired properties. Clearly, in the structure of G165Q the predicted mode of binding of C8 substrate is not affected by the conformation adopted by the Q165 side chain, whereas the longer tails clash with the side chain of Q165, in particular the C10, C11, C12 and C13 atoms. I129 is located at the bottom of the acyl tail binding tunnel. The I129H and I129F point mutation variants were expressed but the in vivo data show that these variants are not active (Fig. 3). Apparently the more polar (I129H) and more bulky, rigid (I129H, I129F) side chains introduce changes with respect to either stability and/or structure, dynamics (or both), that are not compatible with the catalytic function of MECR. The modeling calculations did not allow predicting the structural rearrangements introduced by the G165Q mutation, but nevertheless the availability of a model concerning the mode of binding of the acyl tail in its binding tunnel has been critically important for understanding the properties of the G165Q variant.

Dynamics play an important role in the catalytic cycle of enzymes[30,31]. Notably, despite the crystallographic structures of wild-type MECR and G165Q mutant having the same space group symmetry determined in similar resolution, we observed significantly different B-factors for wild-type and G165Q MECR variant (Supplementary Fig. 6a, b). Firstly, the B-factors are overall significantly higher for the G165Q variant than for wild-type MECR. Also, the magnitude of the differences depends on the regions of the crystallized enzyme, suggesting that some parts of the enzyme may fluctuate relatively more or less depending on the presence of this mutation. The whole catalytic domain and, in particular, the residues of the loop containing Y94, which is known to play a critical role in proton transfer during the

reaction catalyzed by an enzyme homologous to MECR[18], displays much higher B factors in the G165Q mutant than in wild-type, which suggests higher mobility of these regions in the G165Q variant (Supplementary Fig. 6c). Overall, higher fluctuations of the loop near the active site might be related to the observed higher $k_{cat}$ for the C8 substrate in the MECR G165Q mutant than in wild-type MECR (Table 1). The B factors in the loop regions around A133 (shown in Fig. 4d, e) are also much higher in the MECR G165Q variant (Cα B factors around 80–90) than in the wild-type MECR (Cα B factors around 40-50). Therefore, the mutation is likely causing instability in this loop region shaping the shortened substrate binding pocket of MECR G165Q, which possibly explains why the MECR G165Q variant can still catalyze fatty acyl substrates bigger than C8.

Furthermore, we observe a rearrangement of hydrogen bonds of Q165 and surrounding residues in the G165Q mutant when compared with the wild-type MECR (Fig. 4d, e, Supplementary Figs. 7 and 13), associated with changed conformation of the MECR active site due to the mutation. In the wild-type enzyme, the K316 side chain is directly hydrogen bonded to the backbone of G165, while in the mutant it hydrogen bonds to the Q165 side chain amide and the backbone of P130, which may loosen the dynamical coupling between the two enzyme fragments. Also, breaking of hydrogen bonds of S313 (backbone) and N132 (side chain) is observed in the mutant. Finally, the interactions of the highly conserved E107 (Supplementary Table 5) side chain with the highly conserved N83 side chain (located at the base of the catalytic loop containing Y94) and amide hydrogen of the V166 backbone are slightly altered. In the wild-type E107 side-chain carboxylate oxygen atoms form hydrogen bonds with both N83 and V166, while in the G165Q variant a hydrogen bond is only with V166. Overall, these changes in hydrogen-bonding networks may affect structure and dynamics, and further—catalytic function, of the MECR enzyme.

Complete loss of mtFAS does not affect the cellular fatty acid profile and thus there were no detectable changes in fatty acid profile in yeast strain expressing G165Q variant neither (Supplementary Fig. 11). These results were expected because the role of mtFAS in generation of total cellular fatty acid pool in minor compared to cytosolic FAS. We were not able to analyze directly the effect of MECR G165Q mutation on mitochondrial acyl-ACP pool. Instead, we show that the activities of lipoic acid containing enzyme complexes PDH and α-KDH are not affected. Similarly, heme-dependent catalase activity in Δetr1 cells carrying a plasmid expressing MECR G165Q variant was on the same level with wild-type cells. These results indicates that MECR G165Q mutation does not prevent synthesis of short-chain fatty acids up to C8 and the respiratory deficient phenotype of cells carrying this mutation must be due to the lack of long-chain fatty acids.

In yeast Δetr1 strain, the level of Cox1, a mitochondrially encoded subunit of respiratory complex IV, is severely reduced[4]. This is due to a decrease in Cox1 translation caused by a respiratory complex IV assembly defect. We also studied the Cox1 expression level in Δetr1 cells that express G165Q mutant MECR and noticed that Cox1 was undetectable. This is an interesting observation because ACP has been found associated with the mitochondrial ribosome in mammalians and trypanosomas[11,12]. In light of the high conservation of mtFAS-related processes between yeast and mammals and the clear association of mtFAS with the mitochondrial translational processes[4], it is conceivable that acylated ACP may also play a role in ribosomal function in yeast mitochondria. We will address the question of the role of long chain acyl-ACP in mitochondrial translation in future studies, using the tools created for this report.

To conclude, our current work demonstrates the successful engineering of chain-length preference of a lipid metabolizing enzyme towards medium-chain substrates. The engineering was based on the crystal structure of MECR, molecular modeling and in vitro testing. Experiments in vivo showed that both the wild-type MECR as well as the G165Q variant having low catalytic activity towards long-chain acyl-

substrate due to a shortened binding cavity recovered cellular lipoic acid levels of Δetr1 yeast cells. However, only the wild-type MECR, but not the substrate length-restricted MECR G165Q variant was able to support growth of this strain on a non-fermentable carbon source. The results allow the interpretation that mtFAS has a dual role as an integrator in mitochondrial function: to provide long-chain acyl-ACP to maintain mitochondrial respiratory capacity and octanoyl-ACP for lipoic acid synthesis. The MECR G165Q variant now provides a tool to elucidate the regulatory mechanisms of the long-chain acyl-ACP species independent of mtFAS function in lipoylation in further studies.

## Methods

### Computational methods

**Modeling of the initial complex of wild-type MECR with NADP⁺ and substrate.** Modeling of the complexes was performed using the Maestro suite (Schrödinger Release 2017-4: Maestro, Schrödinger, LLC, New York, NY, 2017, http://www.schrodinger.com/). The model was based on the unliganded human MECR crystallographic structure (PDB entry 2VCY[19]). The missing NADP⁺ cofactor and 2E-butenoyl-CoA substrate fragment were aligned from the *Candida tropicalis* Etr1 structure (PDB entry 4WAS[21]) in chain A active site of the modeled MECR dimer. Sequence identity of MECR (UniProt code Q9BV79) and *C. tropicalis* Etr1 (UniProt code Q8WZM3) is 34% as calculated from the alignment with the Clustal Omega program[32] at the UniProt webpage[33] (see also Supplementary Fig. 2). The structural alignment of the human MECR and *C. tropicalis* Etr1 structures (Supplementary Fig. 14) displays distinct conformational differences in the coenzyme A binding site, while active site protein backbone conformations are similar. Therefore, we focused on modeling the MECR complex only with the 2E-alkyl fragments of substrates. In the protonated human MECR enzyme complex, these alkyl fragments of the 2-alkenoyl substrate series (2E-butenoyl-CoA to 2E-hexadecenoyl-CoA) were built incrementally, starting from the crystallized fragment. The two-carbon fragments were subsequently added and minimized in the cavity, beginning with the shortest and ending with the longest alkyl chain, following the shape of the cavity in the enzyme, suggested by the previous mutational study[19]. Based on this modeled complex, positions for the mutations were selected. The coordinates of the initial models for wild-type MECR and selected variants are provided in Supplementary Data 1.

**Building and initial selection of the MECR mutations for experimental evaluation.** Mutants based on the MECR/2E-C16-substrate complex model were built for the selected mutation points. Using Maestro, we computationally substituted the selected positions with amino acid residues, followed by visual inspection of the most frequent rotamers (using Maestro). We evaluated the fit of the particular mutation/rotamer in the protein structure (to minimize clashes) and the propensity of the mutation to block the alkyl chain of the substrate in the correct location (many clashes with ≥C10 substrates, none or few clashes with ≤C8 substrates).

**Residue conservation analysis.** The ConSurf webserver[34–37] (5 Sep 2018) was also used to gain insight into the amino acid conservation patterns in the sequences homologous to MECR with the PDB structure 2VCY[19] (chain A) taken as input. The alignment was performed for the UNIREF90 sequences[38].

**Docking simulations to the wild-type and mutated MECR variant.** For the following computations, the Maestro suite ver. 2019-4 was used. Structure preparation was done with the Protein Preparation Wizard tool[39] and docking simulations were performed with Glide[40–42]. Docking simulations were performed to the prepared wild-type and G165Q MECR structures (dimers) obtained in this work, with NADP⁺ and substrate fragment modeled in the chain A active site, similarly as for the

initially built complex (described above). To avoid docking of very large ligands, which would be difficult even with advanced simulation approaches, the substrates were truncated to fragments containing only 2E-enoyl tails without coenzyme A or ACP, which serve as substrate carriers (Supplementary Fig. 1). The fragments were further prepared using the LigPrep tool and Epik program[43,44] at pH 7.0. In all docking simulations, to be consistent with the initial modeling data, the OPLS2005 force field was used[45,46]. Docking grid size and position were based on the modeled-in 2E-C16 substrate fragment. Docking poses were restrained to the position of the substrate core (defined in Supplementary Fig. 1a) with a standard RMSD threshold of 2.0 Å for the substrate core. This strategy was used because, on one hand, the MECR mutation considered was located far from the ACP or coenzyme A binding site and therefore the assumption that this binding site would not be significantly affected by the mutation (assuming that the MECR variant would overall fold correctly) is reasonable. On the other hand, under these assumptions, the position of the substrate "core" that undergoes the catalyzed reaction is significantly determined by the position of the bound coenzyme A (or ACP) carrier, stabilized by the precise hydrogen-bonding network with the enzyme and NADPH (as, e.g., in the *C. tropicalis* Etr1 structure: PDB entry 4WAS, see Supplementary Fig. 15). The best 10 docking poses were refined, and ten output poses were requested. Additionally, the ligand poses with RMSD over 1.4 Å were discarded in post-processing, to filter out most poses having the "core" moiety with the flipped enoyl group. The SP (standard-precision) semi-rigid docking, and not XP (extra-precision), protocol was used because the latter is more appropriate when the receptor is close to the bound conformation. In this study, the docking receptor is modeled based on the apo-enzyme, and the MECR enzyme conformation near the active site likely differs from the holoenzyme: some conformational changes were observed between the human MECR apoenzyme (PDB entry 2VCY) and homologous holoenzyme *C. tropicalis* Etr1 (PDB entry 4WAS, see Supplementary Fig. 14). These changes could be partially attributed to the sequence differences (Supplementary Fig. 2), but likely also to the apo/holo state of the enzyme. The coordinates for docked substrates in 7AYB and 7AYC structures are available in Supplementary Data 1.

### Cloning and mutagenesis

Cloning of the *MECR*-pYE352 plasmid containing the coding sequence of *S. cerevisiae COQ3* mitochondrial targeting sequence (MTS) and expressing the *H. sapiens MECR* ORF under control of the yeast CTA1 promoter is described earlier[47]. This plasmid was used as a template to generate MECR variants through site-directed mutagenesis by Quick-Change® Site-Directed Mutagenesis kit (Agilent Technologies, Cedar Creek, CA, USA) according to the manufacturer's instructions. The sense and antisense primers used in mutagenesis are listed in Supplementary Table 6. All plasmid constructs and mutations were verified by sequencing performed by the Biocenter Oulu Sequencing Center.

### Yeast strains, media and genetic methods

Wild-type BJ1991α (*MAT α, leu2, trp1, ura3-52, pep4-3, prb1-1122, gal2*)[48] and BJ1991α Δetr1 (*MAT α, leu2, trp1, ura3-52, pep4-3, prb1-1122, gal2; ybr026c::kanMX4*)[14] mutant strains of *S. cerevisiae* are used in this study. Media used are as follows: YPD [1% yeast extract (DIFCO, NJ, USA), 2% Bacto-peptone (DIFCO), 2% D-glucose], SCD [0.67% Yeast Nitrogen Base without amino acids (DIFCO), 0.19% Synthetic complete drop out mix without uracil (Sigma-Aldrich, St. Louis, MO, USA), 0.008% uracil, 2% D-glucose], SCG [0.67% Yeast Nitrogen Base without amino acids (DIFCO), 0.19% Synthetic complete drop out mix without uracil (Sigma-Aldrich), 0.008% uracil, 3% glycerol], SCD/-Uracil [0.67% Yeast Nitrogen Base without amino acids (DIFCO), 0.19% Synthetic complete drop out mix without uracil (Sigma-Aldrich), 2% D-glucose], SCG/-Uracil [0.67% Yeast Nitrogen Base without amino acids (DIFCO),

0.19% Synthetic complete drop out mix without uracil (Sigma-Aldrich), 3% glycerol]. Solid media was prepared with 2% Agar (Fisher BioReagents™, Geel, Belgium).

## Yeast transformations

The plasmids were transformed into the BJ1991 α wild-type and Δ*etr1* yeast strain by using one-step yeast transformation protocol[49].

## Yeast respiratory growth assay/spotting assay

The yeast strains BJ1991 α containing empty expression plasmid pYE352; BJ1991 α Δ*etr1* containing empty expression plasmid pYE352; BJ1991 α Δ*etr1* carrying *MECR* -pYE352 wild-type or mutant plasmids were grown in synthetic complete media with glucose lacking uracil (SCD-URA) overnight (16 h). Overnight yeast cultures were used to inoculate fresh SCD-URA culture to an optical density of 0.1 and grown for about 4 h. Cells were harvested and adjusted to $OD_{600}$ of 0.5. Serial dilutions of undiluted, 1:10, 1:100 and 1:1000 was made and 2 μl of each dilution of the cells were spotted on SCD-URA, SCG-URA and YPD. The plates were grown at +30 °C for 4 days. The mutant samples were spotted on two identical plates poured from the same medium preparation and both plates contained all controls (BJ1991 α containing empty expression plasmid pYE352; BJ1991 α Δ*etr1* containing empty expression plasmid pYE352 and BJ1991 α Δ*etr1* carrying *MECR* -pYE352 wild-type). To analyze the loss of mitochondrial DNA (ρ°), BJ1991 α Δ*etr1* carrying *MECR* -pYE352 wild-type and BJ1991 α Δ*etr1* carrying *MECR* -pYE352 mutant G165Q plasmids were transformed with yeast wild-type *Etr1* plasmid[14]. These strains were tested for respiratory growth on fermentable (SCD) and non-fermentable (SCG) growth media. The plates are grown for 4 days at +30 °C.

## Protein isolation from yeast strains

Yeast proteins were isolated according to the modified protocol from Platta et al. 2004[50] via trichloroacetic acid (TCA) precipitation. For the determination of lipoylation and MECR expression level in BJ1991 α containing empty expression plasmid pYE352; BJ1991 α Δ*etr1* containing empty expression plasmid pYE352; BJ1991 α Δ*etr1* carrying *MECR* -pYE352 wild-type or mutant plasmids were grown in SCD-URA medium at +30 °C for about 16 h. Then overnight cultures were transferred to 25 ml SC-glycerol-URA media containing 0.05% glucose and cultured until the $OD_{600}$ reached ~4–5. After reaching the desired OD, the cultures were harvested by centrifugation at 3000 x *g* for 5 min at + 4 °C. The cells were washed with 50 ml sterile water and centrifuged again at 3000 x g for 5 min at +4 °C. The cells were suspended to 1 ml sterile and centrifuged at full 20,800 x *g* for 10 s. The wet weight of the cells was determined and 1 ml of sterile water was added to 100 mg of cells. 300 μl of cell suspension transferred to Eppendorf tubes and 15 μl of 1 M potassium phosphate buffer (pH 7.4) was added. 100 μl of 50% TCA was added and mixed well. Cells were incubated at −70 °C for 30 min, thawed out on ice and centrifuged at 20,800 x g for 10 min. After discarding the supernatant, 500 μl of ice-cold 80% acetone was added to wash the lysate. The lysate was centrifuged for 5 min at 20,800 x *g* and the resulted pellet was re-suspended into 60 μl of freshly prepared 1% SDS/ 0.1 M NaOH solution.

## Protein isolation from yeast mitochondria

Mitochondria isolation was performed according to Meisinger et al. 2000[51] to analyze the steady state level of mitochondrial respiratory chain complex protein Cox1. Yeast strain BJ1991 α containing empty expression plasmid pYE352; BJ1991 α Δ*etr1* containing empty expression plasmid pYE352; BJ1991 α Δ*etr1* carrying *MECR* -pYE352 wild-type or mutant plasmids were grown in SCD-URA medium at +30 °C for about 16 h. Then overnight cultures were transferred to 50 ml SCD-URA media and cultured for ~24 h followed by a 400 ml culture ~24 h until the $OD_{600}$ reached ~5. The cultures were harvested by centrifugation at 3000 x g for 5 min at room temperature. The cells were washed with 25 ml sterile water and centrifuged again at 3000 x g for 5 min at room temperature before they were resuspended in 2 mL DTT buffer (100 mM Tris-H$_2$SO$_4$, pH 9.4, 10 mM DTT) per gram of cells and shaken slowly (-100 rpm) at +30 °C for 20 min. The cells are centrifuged again at 3000 x g for 5 min at room temperature and washed with 25 mL of zymolyase buffer (1.2 M sorbitol, 20 mM potassium phosphate buffer pH 7.4) before centrifuged again at 3000 x g for 5 min at room temperature. Each gram of cell pellets was resuspended in 7 ml zymolyase buffer containing 5 mg of zymolyase 20 T (MP Biomedical, Irvine, CA, USA) and incubated at slow speed (-100 rpm) at +30 °C for 45 min. Homogenization was carried out by 15 strokes in a glass-Teflon potter in 30 mL ice-cold homogenization buffer (0.6 M sorbitol, 10 mM Tris-HCl buffer pH 7.4, 1 mM EDTA, 1 mM PMSF, 0.2% (w/v) BSA). The homogenate was centrifuged at 1500 x *g* for 5 min at +4 °C and the organelles of the supernatant were collected. The supernatant was centrifuged at 3000 x *g* for 5 min at +4 °C. Then, crude mitochondria were pelleted from supernatant by centrifugation at 12,000 x *g* for 15 min at +4 °C. Mitochondrial pellet was washed for three times with 25 ml ice cold SEM buffer (250 mM sucrose, 1 mM EDTA, 10 mM Mops, pH 7.2) by centrifugation at 12,000 x *g* for 15 min at +4 °C. Resulted mitochondrial pellet was re-suspended in SEM buffer.

## Western blot analysis

Protein samples extracted from the yeast cells (by TCA precipitation) and crude mitochondrial proteins were separated on SDS-PAGE. Equal amounts of protein from TCA precipitation were loaded on 12% SDS-PAGE gel or 17 μg crude mitochondrial proteins were loaded on 4-20% 10-well Mini-PROTEAN® TGX™ precast gels (Bio-Rad Laboratories, Hercules, CA, USA). The proteins were transferred onto 0.2 μm nitrocellulose membrane (Trans-Blot Turbo Transfer pack, Bio-Rad) using Trans-Blot Turbo Transfer System (Bio-Rad). The membranes were blocked using 5% skim milk in 20 mM TRIS base, 137 mM NaCl, pH 7.6 with HCl, 0.1% Tween20 (TBST) solution. Polyclonal rabbit anti-lipoic acid antiserum (EMD Millipore Corporation, USA; 437695; RRID AB_212120; 1:2500) was used for detection of lipoylated proteins, anti-Mecr antibody (ProteinTech Group, USA; 51027-2-AP; RRID: AB_10863346; 1:3000) to detect MECR and mouse monoclonal anti-MTCO1 antibody (anti-COX1) (Abcam, UK; ab110270; RRID: AB_10863346; 1:333) was used to detect the levels of Cox1. Ponceau staining, β-actin (Abcam; [mAbcam8224]; ab8224; RRID: AB_449644, 1:3000) and anti-VDAC1/Porin (Abcam, UK; [16G9E6BC4]; ab110326 RRID: AB_10865182, 1:1000) immunodetection were used as loading controls. Secondary antibodies were anti-mouse IgG (H + L), HRP conjugate (Promega, USA; W4021; RRID AB_430834, 1:2500), or Immun-Star goat anti-rabbit (GAR)-HRP conjugate antibody (Bio-Rad; 170-5046; clone number 430RRID: AB_11125757, 1:2500). Antibody detection reaction was developed by using a Clarity™ western ECL substrate (Bio-Rad), Molecular Imager ChemiDoc™ XRS + equipment and Image Lab version 3.0 software (Bio-Rad).

## Catalase, PDH, α-KDH and fumarase activity assays

For enzyme activity assays, yeast cells were grown as described in mitochondria isolation procedure above. For catalase activity assay, 2 ml of spheroplasts were harvested after addition of homogenization buffer (0.6 M sorbitol, 10 mM Tris-HCl buffer pH 7.4, 1 mM EDTA, 1 mM PMSF, 0.2% (w/v) BSA). The catalase activity was measured from yeast spheroplast as described[52]. Mitochondrial fractions for PDH, α-KDH and fumarase activity were prepared as same manner as described in Schonauer et al. 2009[53]. Mitochondrial pellets were suspended in breaking buffer (0.1 M potassium phosphate buffer pH 7.4, 4 mM EDTA, 10 μM thiamine pyrophosphate, 0.2 mM phenylmethanesulfonyl fluoride, 0.68 mg liter$^{-1}$ pepstatin A, cOmplete™ EDTA-free Protease Inhibitor Cocktail (Roche)) followed by the addition of Triton X-100 to a final concentration of 1%. Particulate matter was removed by

centrifugation for 20 min at 37,000 x g. PDH and α-KDH activity was assessed by following the formation of NADH at 340 nm as described[53]. Fumarase activities were measured as described[54]. Protein concentrations in spheroplast and mitochondrial extracts were estimated by Bradford assay (BioRad) by using bovine serum albumin (Sigma) as a standard.

## Total fatty acid analysis from yeast cells

Lipids and fatty acids from $100 \times 10^6$ yeast cells were extracted by adding 170 μl 0,1 M HCl, 280 μl chloroform/methanol (50/50, v/v) and 20 μl of 10 μM [2H31] hexadecanoic acid as an internal standard. After centrifugation, upper phase was extracted again with 300 μl of CHCl3/MeOH/H2O (70/40/10, v/v/v) and lower phases were combined. 100 μl of combined lower phases was evaporated under a stream of nitrogen gas at 50 °C to dryness. The dried lipid pellet was redissolved in 180 μl methanol. After addition of 20 μl 3 M KOH (in water), the resulting solution was heated to 80 °C for 2 h. Unpolar lipids were removed by extraction with twice 500 μl hexane. After acidification with 50 μl formic acid, free fatty acids were extracted twice with 500 μl hexane each. The combined fatty acid extracts were evaporated to dryness under a stream of nitrogen at 50 °C. The dry sample extracts were redissolved in 50 μl iPrOH.

For LC/MS analysis, 3 μl of sample was applied to the Accucore Biphenyl column (2.6 μm particles, 100 × 2.1 mm) (Thermo Scientific, Bremen, Germany) at 30 °C. Fatty acids were separated with mobile-phase buffer A containing 5 mM NH4OAc in acetonitrile/water (95/5, v/v), and solvent B consisting of 5 mM NH4OAc in acetonitrile/water (95/5, v/v). After sample application, the LC gradient program was 0% solvent B for 2 min, followed by a linear increase to 100% solvent B within 8 min, then maintaining 100% B for 9 min. The flow rate was maintained at 200 μL/min. The eluent was directed to the ESI source of the QE-MS from 2.0 min to 13.0 min after sample injection. MS analysis was performed on a Q-Exactive mass spectrometer (Thermo Scientific) applying the following scan and HESI source parameters: scan ranges: 150–500 m/z (negative mode); resolution: 70,000; AGC-Target 1E6; maximum injection time: 200 ms; sheath gas: 20; auxiliary gas: 1; sweep gas: 0; aux gas heater temperature: 120 °C; spray voltage: 3.0 kV; capillary temperature: 320 °C; S-lens RF level: 50.0. Signal determination and quantitation were performed using TraceFinder V3.3 software (Thermo Fisher).

## *Escherichia coli* strains

The *E. coli* strain TOP10F' (F' [*lac*Iq, *Tn*10(Tet R)] *mcr*A Δ(*mrr-hsd*RMS-*mcr*BC) φ80*lacZ*ΔM15 Δ*lac*X74 *rec*A1 *ara*D139 Δ(*ara-leu*)7697 *gal*U *gal*K *rps*L(Str R) *end*A1 *nup*G) was used for plasmid cloning and propagation (Invitrogen, Carlsbad, CA, USA). *E. coli* BL 21 Star (DE3) pLysS strain was used to overexpress and purify wild-type and mutant protein for kinetic assay and crystallization screen.

## Cloning, overexpression and protein purification

*MECR- pET23a* plasmid construct encoding wild-type MECR or MECR mutants without the mitochondrial targeting sequence and with C-terminal His-Tag was generated as described in Miinalainen et al. 2003[15]. *E. coli* BL 21 Star (DE3) pLysS strain was transformed with these plasmids for overexpression and cultured in LB medium containing ampicillin and chloramphenicol at +37 °C. Expression of MECR wild-type and mutant proteins were induced by 0.4 mM isopropyl-β-D-thiogalactoside (IPTG) at OD600 0.6–0.7 for overnight at +37 °C.

Cells expressing the MECR wild-type and mutant protein were harvested and resuspended in the binding buffer (5 mM imidazole, 500 mM NaCl, and 20 mM Tris-HCl, pH 7.9) and sonicated. After centrifugation, the supernatant was mixed with Ni-NTA Superflow matrix (Qiagen, Hilden, Germany) and proteins containing a His-Tag were bound to the matrix in +4 °C for 1 h. Unbound proteins were washed with binding buffer followed by washing buffer (32.5 mM imidazole,

500 mM NaCl, and 20 mM Tris-HCl, pH 7.9). The protein was eluted with linear imidazole gradient from 32.5 mM to 500 mM and the fractions containing wild-type MECR or MECR G165Q mutant with His-Tag were collected and pooled. The pooled fractions were concentrated and applied to size-exclusion chromatography column (Superdex TM 200 10/300, GE Healthcare, Uppsala, Sweden). Buffer containing 100 mM sodium phosphate, 150 mM NaCl, 1 mM EDTA, and 1 mM NaN₃, pH 7.4 and 10% glycerol was used as an eluate. This buffer is also the protein storage buffer. The peak fractions were combined, concentrated and stored at −70 °C until used. Pure protein band was confirmed by SDS-PAGE analysis.

## Determination of kinetic parameters for MECR WT and MECR G165Q

The *2E*-enoyl-CoA reductase activity of wild-type and G165Q MECR variant were determined spectrophotometrically by using a JASCO V-660 spectrophotometer with JASCO Spectra Manager Software. The reaction mixture contained 125 μM NADPH, 100 μg of bovine serum albumin in 50 mM potassium phosphate, pH 7.5, and either wild-type or G165Q mutant MECR. The reactions were initiated by adding the substrate to a final volume of 500 μl at 25 °C. The used substrates were: 2*E*-hexenoyl-CoA, 2*E*-octenoyl-CoA, 2*E*-decenoyl-CoA, 2*E*-dodecenoyl-CoA, 2*E*-tetradecenoyl-CoA and 2*E*-hexadecenoyl-CoA at the final concentrations of 1.25–180 μM. The kinetic data were fitted to the Michaelis-Menten plot by using GraphPad Prism computer software and catalytic turnover numbers were calculated as described as Miinalainen et al. 2003[15].

## Circular dichroism (CD) spectroscopy

The CD spectra of the purified proteins were obtained using a a Chirascan™ CD spectrophotometer (Applied Photophysics, Surrey, U.K.) in a quartz cuvette with 1 mm path length. Before the experiments, protein buffer was exchanged into 10 mM potassium phosphate buffer, pH 7.6. During the experiment protein concentrations were 0.34 mg/ml and 0.21 mg/ml for wild-type MECR and MECR G165Q mutant, respectively. For the determination of the $T_m$, the sample was heated at a rate of 1 °C per minute from +22 to +94 °C. Data analysis was carried out using Pro-Data Viewer (Applied Photophysics), CDNN (http://bioinformatik.biochemtech.uni-halle.de/cdnn) and Global3 (Applied Photophysics).

## Crystallographic studies of wild-type MECR and MECR G165Q variant

Crystallization trials were carried out in the crystallization facility of the Biocenter Oulu Structural Biology core facility by using the sitting drop vapor diffusion method. Initial screening was done with the wild-type MECR (19.5 mg/ml, in 100 mM sodium phosphate,150 mM NaCl, 1 mM EDTA, 1 mM NaN₃, pH 7.4 and 10% glycerol) by using the in-house screen[55] and IQ 96-well sitting drop plates (SPT Labtech, Melbourn, Hertfordshire, U.K.) using the Mosquito LCP nanodispenser (SPT Labtech) at two different temperatures (at +4 °C and +22 °C). The plates were imaged by using Formulatrix Rock Imagers RI54, and the crystallization results were monitored with the IceBear software expert system[56]. MECR crystals were obtained at +4 °C directly form the in-house screen and optimized in 16.82% PEG 6000, 1 M NaCl, 100 mM acetic acid, pH 4.5. The volume of the crystallization drop was 300 nl and the drop ratio was 1:2 for the reservoir and protein sample, respectively. The wild-type MECR crystals appeared in 20 h and grew to their final size in 5 days. The MECR G165Q variant (8.7 mg/ml) was crystalized in the same condition, also at +4 °C, as used for the wild-type. The crystals appeared in 2 days and grew to the final size in 5 days.

For the X-ray experiments, crystals were cryocooled in the cold room by transfer into liquid nitrogen. Before cryocooling, the crystals were incubated few seconds in the cryosolution, which was made by

mixing 7 µl of the well solution and 3 µl of 100% glycerol for all protein crystals. From both crystals data sets with a resolution of 2.2 Å (wild-type MECR) or 2.5 Å (MECR G165Q) were obtained, using the in-house Microstar X8 Proteum Cu-rotating anode X-ray generator (Bruker) of the Biocenter Oulu Structural Biology core facility. A synchrotron source was also used for data collection and higher resolution data sets were obtained at the Diamond Light Source (DLS, Didcot, United Kingdom) beamline I04 (for wild-type MECR 1,85 Å and for MECR G165Q mutant 2,02 Å). The home source data were integrated and processed using the Proteum2 software package (Bruker), whereas the synchrotron data were integrated with DLS autoprocessing pipelines based on XDS (wild-type MECR) or DIALS (G165Q variant)[57,58]. All data were scaled with AIMLESS[59].

Previously determined MECR structure (PDB entry 2VCY[19]) was used as an initial search model in the molecular replacement calculations of wild-type MECR by PHASER and by using the 2.2 Å data collected by using the in-house diffraction data[60]. The structure of wild-type MECR was further used as a model to solve the structure of the mutated variant by using the in-house collected X-ray data. However, the structure refinement and model building were performed by using the synchrotron collected data sets. Refinement and model validation were done using the Phenix package[61] and the model building was done with COOT[62]. The final refinements excluding model building were performed with PDB-REDO web server[63]. A summary of the refinement and validation results are presented in Supplementary Table 3. The coordinates and structure factors of the refined structures have been deposited in the Protein Data Bank with accession codes 7AYB (wild-type MECR) and 7AYC (MECR G165Q variant).

### Statistical analysis
Results are presented as mean ± standard deviation (SD). Data were analyzed by Student's *t*-test (two-tailed) and *P*-values <05 were considered significant.

### Reporting summary
Further information on research design is available in the Nature Portfolio Reporting Summary linked to this article.

## Data availability
Atomic coordinates and structure factors of wild-type and G165Q MECR mutant have been deposited to the Protein Data Bank under accession number 7AYB (wild-type MECR) and 7AYC (MECR G165Q variant). Raw diffraction images are available at IDA (https://doi.org/10.23729/aa2ec443-83bb-4b68-a02d-4d3fc7bc0fbd). Previously published crystal structures used to derive the models shown are 2VCY[19], 1ZSY and 4WAS[21]. The authors declare that all other data supporting the findings of this study are available within the paper and its supplementary information, supplementary data 1 and source data files. Source data are provided with this paper.

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

## Acknowledgements

This work was supported by grants from the Academy of Finland [314925, A.J.K], the Sigrid Juselius Foundation (J.K.H, A.J.K, K.J.A), the Mary and Georg C. Ehrnrooth Foundation [202100005, K.J.A], Jane and Aatos Erkko Foundation (A.J.K) and Finnish Cultural Foundation [00220849, M.T.R.]. We thank Lamiya Jabiyeva for the technical help in PDH, α-KDH, fumarase and catalase activity assays. Biocenter Oulu sequence center and the use of the facilities and expertise of the Biocenter Oulu Structural Biology core facility, a member of Biocenter Finland, Instruct-ERIC Centre Finland and FINStruct, as well as the use of the facilities and expertise of the Biocenter Oulu biophysical protein analysis core facility, a member of Biocenter Finland, are gratefully acknowledged. J.P.-H. and R.C.W acknowledge the support of the Klaus Tschira Foundation. J.P.-H. was additionally supported by Polish National Science Centre [2016/21/D/NZ1/02806, J.P.-H.] and by BIOMS program at the Interdisciplinary Center for Scientific Computing IWR, University of Heidelberg.

## Author contributions

M.T.R, A.J.K., J.K.H and K.J.A. designed the experiments. M.T.R. performed the yeast and protein experiments and analyzed the data. J.P.-H.

and R.C.W. designed the computational analysis and J.P.-H. carried out the computational modeling and simulation. M.K.K. and R.K.W. supervised the crystallization studies. W.S. produced the substrates for enzyme kinetic measurements. M.T.R., J.K.H. and K.J.A. wrote the paper with input from all the other authors.

## Competing interests

All the authors declare no competing interests
