## [Peer Review File · Nature Communications]

REVIEWER COMMENTS

Reviewer #1 (Remarks to the Author):

The paper of Autio and co-workers reports an exciting computational-experimental work that tailors the active site of an enzyme MECR reductase through rational computational methods for accepting only desired (short-chain) substrates. The modelled enzymes are tested, and one is confined to work as expected. The mutant is used to derive important insights into the requirements for long-chain acyl-ACPs for mitochondrial respiration.

In general, the work is very sound, the conclusions have a high impact in the field, and the data provides solid support to the findings. I liked to read it and believe it merits publication in Nat. Communications. Nevertheless, I have a few suggestions that should help to strengthen the work, which I list below:

1. Computational chemists are thrilled with the perspective of creating enzymes by design. Unfortunately, the field is plagued by false positives. Therefore, understanding failure is key for evolution. During the in vitro screening of the MECR mutants, all but one mutant presented the desired properties, the others have failed for different reasons. I understood that some of the mutations still processed long-chain while others did not accept even small chains, below 8 carbon atoms. It would be very important for the field of enzyme design and evolution if the authors tried to explain these results, at least for some of the mutations, to understand why the initial designs failed to reach the objectives.

2. A similar line of thought applies to the kinetic analysis. Increasing K_{cat} through engineering is a very hot topic, and thus it would be important to try to understand the underlying molecular factors for an increase or decrease in reaction rate.

I guess it has to do with the positioning of the core in relation to the catalytic residues. the discussion should be rooted on accurate stereo requirements for reactivity. A recent publication in Chem. Rev. (<https://doi.org/10.1021/acs.chemrev.1c00147>) summarizes these requirements precisely for the enzyme under study and might help to cement the discussion.

3. A final aspect that merits discussion are the docking models. I was not totally convinced that constraining the docking for the core atoms to match was the best option. Does this constraint generate poses which are not the real ones, the ones that really happen in the mutated enzyme? This aspect should be clarified.

It would be great if the authors also superimpose the poses of the same substrates in the wild-type and mutant enzymes in a SI panel.

Reviewer #2 (Remarks to the Author):

The manuscript addresses an interesting unresolved question of mitochondrial biology: Is de novo synthesis of long chain fatty acids (longer than the C8 needed for lipoyl synthesis) within mitochondria essential for proper mitochondrial function or is this capability of the mtFAS II system dispensable? The authors present an interesting approach combining theoretical work with mutagenesis, structural biology, as well as in vivo and in vitro analyses. First, a mutant MECR enzyme with a shortened fatty acid binding pocket is predicted and designed to putatively being unable to synthesize longer fatty acid chains. Out of several possibilities the authors select a MECR mutant G165Q that is able to (partially) restore lipoylation but not growth of yeast *etr1* (ortholog of human MECR) deletion cells. This result suggested that the mutant is unable to handle longer chains. However, enzymological analysis supports this idea only partially, as the mutant is only partially (6-10-fold) impaired in converting longer fatty acyl chains compared to the wild-type enzyme, but is not inactive as expected from the in vivo results. Could it be that the mutant simply synthesizes octanoyl (for lipoate) too slowly to support normal growth. To clarify this apparent discrepancy, an essential experimental information would be the measurement of the chain lengths synthesized by the mutant enzyme in vivo (analyzed as by Angerer et al.). Further, a simple further experiment is the measurement of the PDH and KGDH activities in yeast cell extracts. The authors then crystallized the wild-type and mutant MECR proteins, and get structures similar to those of previously resolved members of this protein family. Comparison of wild-type and G165Q mutant structures shows differences in hydrogen bonding as well differences to the predicted modelled structure. Residues around the active center of MECR G165Q display much higher B factors than in wild-type suggesting higher mobility of these regions in variant. Since this made interpretations a bit weak, the authors then perform theoretical docking studies of different C chain length substrates to the crystallized proteins. In my view this approach does not convincingly replace the in vivo analysis of the mutant enzymes' capability to synthesize various carbon chain lengths. Finally, the authors try to find a molecular explanation for the proposed inability of mutated MECR to synthesize longer chain fatty acyl groups, i.e for the putative role of longer C chains in mitochondrial physiology, the key question of this paper. They show that the MECR G165Q mutant lacks detectable amounts of Cox1. This is indeed an interesting finding, yet raises the question whether this is due to a transcriptional effect (as seen in Ref. 4). Moreover, reading ref. 4, this leaves me puzzled why in this earlier work Cox1 protein is only slightly diminished (particularly after de novo protein synthesis) while in their new work the authors see a complete absence of Cox1 in the deletion and mutant strains. What about other mitochondrially encoded proteins? What about protein synthesis activity in these cells (as done in Ref. 4). Overall, this is an interesting study, yet the lack of convincing data that longer carbon chain fatty acids are absent in the MECR mutant and why longer carbon chains may be needed to sustain respiratory growth of the mutant yeast cells still limits the impact of the study.

Specific comments

1. Introduction: Please define MDR (to avoid confusion with multidrug resistance).
2. Results start with a kind of summary what will be done. Is this really needed?
3. Fig. 3: Lipoylation of Lat1 in part b is hardly detectable. Why? Please provide longer exposures. In a the panels seem to have been assembled from separate plates. This is not indicated properly.
4. Are the MECR G165Q cells rho zero or rho plus? If the first is true, this would explain why Cox1 is undetectable. A mitochondrial protein biosynthesis assay would be excellent to know what the effects in the various cells are.
5. Fig. 4 (in my view) is more a supplement figure.
6. Fig. 5 and 6 can easily be combined because they address the same issue.
7. The paragraph on heme in Discussion is not really clear to me. Either this topic needs better explanation (what is the link of KGDH and heme) or - even better – heme content is measured (cytochromes, catalase).

Reviewer #3 (Remarks to the Author):

The manuscript entitled “An engineered variant of MECR reductase reveals indispensability of long-chain acyl-ACPs for mitochondrial respiration,” by Rahman, et al., offers an exploration into the mitochondrial FAS enoylreductase MECR. At its core, the manuscript attempts to better understand how MECR contributes to fatty acid biosynthesis and the metabolic requirements for both lipoic acid precursors and long chain fatty acids. While it has long been dogma that the mitochondrial FAS only produces octanoic acid to supply precursor for lipoylation, recent x-ray and cryoEM structures of other metabolic components, including the ribosome and iron-sulfur biosynthesis among others, indicate that long-chain fatty acids attached to the ACP participate in metabolic regulation. This manuscript sets out to clarify this fundamental contradiction.

The team used heterologous human MECR in a *Saccharomyces cerevisiae* Δ etr1 yeast strain lacking native mtFAS enoyl reductase. Using in silico design of a mutant MECR suitable for octanoate biosynthesis, but unable to reduce longer chain fatty acids, the transgenic strain is used to evaluate the cellular effect of lacking long-chain fatty acids in the presence of endogenous lipoic acid, and thus the effect upon mitochondrial respiration with the presence of lipoylation. The authors claim to combine in silico, in vitro, and in vivo work to compare the stability and function of wild-type MECR and mutant MECR proteins.

However, the manuscript primarily details the computational studies in development and x-ray crystal structure data collected with the mutant MECR. The in vivo portions of the work are limited to evaluation of mutants expressed in Δ etr1 strain, along with anti-lipoic acid western blots and evaluation of growth on selected media. While these studies do indicate that the mutants are capable of producing

octanoate for lipoic acid, the conclusion that they cannot form longer chain fatty acids is not fully established. More experiments are required to reach this conclusion. Overall, this manuscript addresses a central topic in our understanding of mtFAS that has major implications in mitochondrial metabolism and disease.

In addition to the above, the following items were identified that should be addressed:

1. The sentence “The structurally well characterized...” (Lines 35-37) lacks some transfer from last sentence introducing mtFAS.
2. In Line 59, the word “adjust” seems not quite appropriate- might be better to be changed to “regulate”.
3. There’s an extra dot in the annotation “(Figure 1a.)” (Line 61).
4. In the sentence “We have previously...” (Lines 118-121), the previous studies on three different mutations is quoted, but only two of them was mentioned in the subsequent discussion. Is there any specific reason F324Y is not included in this study?
5. After the sentence “The backbone geometry...” (Lines 123-124), you may also consider mentioning the feasibility to work on I129.
6. A space missing in “glycerol,medium (SCG)” (Line 142).
7. After the sentence “The Km values...” (Lines 173-175), it would be better to have one sentence conclude the difference between the wild-type (decreases to 1/6) and mutant MECR (decreases to 1/14). Also, an enzyme with a lower Km has a higher affinity for its substrate. It’s not clear here why both types of MECR have a higher affinity towards C10 than C6, which may need to be explained explicitly here.
8. Sentences from Line 223-231 and Lines 390-402 analyze the possible effect of hydrogen-bonding shift observed from crystallography. This would be more convincing if some relevant computational work be done to support authors’ hypotheses.
9. In Line 233, the term “apo” is used for the first time without explanation.
10. The paragraph “In previous studies...” (Line 404-410) seems not very logical. Admittedly, the lack of longer-chain fatty acids is probably the direct cause of the respiratory-defect phenotype, but we can’t rule out the possibility that those yeast strains also cannot produce enough C8 precursor which is the prerequisite for longer-chain fatty acids. Also in Line 406, the term “ Δ mtFAS” should be “ Δ etr1”.
11. Lines 416-417, “due to a shortened binding cavity recovered cellular lipoic acid levels of Δ etr1 yeast cells.” This was not experimentally determined. Although lipoic acid was detected, the inability to prepare longer chains was not reported. More experiments are needed to evaluate exactly what fatty acids are made, and which are not. At a minimum, a global FAME analysis of the fatty acids in each strain in various culture conditions would prove valuable.
12. The legend “MECR accepts...” (Lines 867-868) contains key information that should be provided in the main text. In the previous discussion, the object changed from acyl-ACP to acyl-CoA without any transfer, thus being confusing.

13. In Line 896, the format of "(PDB-ID 7AYC)" is not consistent to other "PDB entry".

REVIEWER COMMENTS

Reviewer #1 (Remarks to the Author):

The paper of Autio and co-workers reports an exciting computational-experimental work that tailors the active site of an enzyme MECR reductase through rational computational methods for accepting only desired (short-chain) substrates. The modelled enzymes are tested, and one is confined to work as expected. The mutant is used to derive important insights into the requirements for long-chain acyl-ACPs for mitochondrial respiration.

In general, the work is very sound, the conclusions have a high impact in the field, and the data provides solid support to the findings. I liked to read it and believe it merits publication in Nat. Communications. Nevertheless, I have a few suggestions that should help to strengthen the work, which I list below:

1. Computational chemists are thrilled with the perspective of creating enzymes by design. Unfortunately, the field is plagued by false positives. Therefore, understanding failure is key for evolution. During the in vitro screening of the MECR mutants, all but one mutant presented the desired properties, the others have failed for different reasons. I understood that some of the mutations still processed long-chain while others did not accept even small chains, below 8 carbon atoms. It would be very important for the field of enzyme design and evolution if the authors tried to explain these results, at least for some of the mutations, to understand why the initial designs failed to reach the objectives.

Response: The reviewer is correct, however for an extensive evaluation it is essential to have structures. Currently, a structure is available only for the G165Q variant and we have carefully analyzed this structure. The Q165 side chain is well defined in the electron density map and we think that the hydrogen bond properties of the Q165 side chain are probably critical (which are different/lacking in the other probed point mutation variants). Concerning I129: this residue is deeply buried and therefore replacement by bulky rigid side chains is apparently not tolerated. A paragraph concerning the discussion about the mutations that failed to shorten the acyl binding cavity is added to the Discussion (pages 10-11).

2. A similar line of thought applies to the kinetic analysis. Increasing K_{cat} through engineering is a very hot topic, and thus it would be important to try to understand the underlying molecular factors for an increase or decrease in reaction rate. I guess it has to do with the positioning of the core in relation to the catalytic residues. the

discussion should be rooted on accurate stereo requirements for reactivity. A recent publication in Chem. Rev. (<https://doi.org/10.1021/acs.chemrev.1c00147>) summarizes these requirements precisely for the enzyme under study and might help to cement the discussion.

Response: Thank you for this comment. It is not straightforward to correlate k_{cat} values with molecular properties, in particular for enzymes that have a complicated catalytic cycle, like ETR, for at least two reasons, being that (i) k_{cat} is an apparent rate, being dependent on the rates of the various steps of the catalytic cycle and (ii), more importantly, k_{cat} concerns the rate limiting step of the overall reaction, which very often is not the chemical step, but can be a conformational switch, a binding event, or a product release event. The latter is a well-known phenomenon for NAD/NADP dependent enzymes. We studied the interesting recent review (as mentioned above), which addresses the chemical mechanism of enoyl reductase of cytosolic animal fatty acid synthase in detail. Our manuscript does not address the reaction mechanism but in the updated manuscript we now refer to this information in the Introduction (page 3). When presenting the k_{cat} and K_m values we now point out the uncertainties concerning the molecular interpretation of these values (page 6).

3. A final aspect that merits discussion are the docking models. I was not totally convinced that constraining the docking for the core atoms to match was the best option. Does this constraint generate poses which are not the real ones, the ones that really happen in the mutated enzyme? This aspect should be clarified. It would be great if the authors also superimpose the poses of the same substrates in the wild-type and mutant enzymes in a SI panel.

Response: We thank the Reviewer for this comment, this indeed was not properly explained in the manuscript. The point is that the full substrate for MECR, used in experiments, is a relatively large and flexible *2E*-enoyl-coenzyme A (CoA). Docking of this ligand to human MECR would be rather a difficult task, due to the large size and the fact that we created a model of MECR based on the unliganded structure (PDB: 2VCY), since we had only such a human structure available. If one compares the human MECR and the *C. tropicalis* Etr1 structures (from which we took the substrate conformations missing in the human MECR structure, PDB: 4WAS), one can see that at least one loop close to the ACP/CoA binding sites (residues 287-291 of MECR, new Supplementary figure 13) has quite different conformations in the substrate bound and unbound enzyme states, while the conformations are more similar in the catalytic site. Therefore, it would be hard to dock full ligands to our MECR model.

Furthermore, CoA is stabilized at the entrance of the pocket by the network of hydrogen bonds that precisely position the *2E*-enoyl moiety (that is the actual substrate of the catalyzed reaction) in the active site (this can be seen for example in the structure PDB: 4WAS, new Supplementary figure 14). Therefore, since the mutation sites we considered were locating in the MECR structure near fragments that were close to the *2E*-enoyl tails and far from the CoA binding site, we assumed that the mutations would not significantly affect the interactions of CoA with MECR. Thus, we assumed that CoA serves as a natural constraint for the *2E*-enoyl tails, and the "core" fragment that undergoes the catalytic reaction would be similarly positioned in all the enzyme-substrate complexes as in the *C. tropicalis* Etr1 crystal structure (if the whole substrate is able to bind to the enzyme, and the

catalytic process takes place). Moreover, the docking protocol allowed for some conformational freedom of the substrate "core" (an RMSD [a standard measure of the distance between the two conformations of a particular set of atoms] of 2.0 Å was allowed with respect to the crystallographic position of the "core").

The text has been modified accordingly in the Methods section and two new Supplementary figures 13 and 14 are added to explain this topic. We added the figure (Supplementary figure 9) with superimposed poses of the same substrates to the Supplementary Information.

Reviewer #2 (Remarks to the Author):

The manuscript addresses an interesting unresolved question of mitochondrial biology: Is de novo synthesis of long chain fatty acids (longer than the C8 needed for lipoyl synthesis) within mitochondria essential for proper mitochondrial function or is this capability of the mtFAS II system dispensable? The authors present an interesting approach combining theoretical work with mutagenesis, structural biology, as well as in vivo and in vitro analyses. First, a mutant MECR enzyme with a shortened fatty acid binding pocket is predicted and designed to putatively being unable to synthesize longer fatty acid chains. Out of several possibilities the authors select a MECR mutant G165Q that is able to (partially) restore lipoylation but not growth of yeast *etr1* (ortholog of human MECR) deletion cells. This result suggested that the mutant is unable to handle longer chains. However, enzymological analysis supports this idea only partially, as the mutant is only partially (6-10-fold) impaired in converting longer fatty acyl chains compared to the wild-type enzyme, but is not inactive as expected from the in vivo results. Could it be that the mutant simply synthesizes octanoyl (for lipoate) too slowly to support normal growth. To clarify this apparent discrepancy, an essential experimental information would be the measurement of the chain lengths synthesized by the mutant enzyme in vivo (analyzed as by Angerer et al.). Further, a simple further experiment is the measurement of the PDH and KGDH activities in yeast cell extracts. The authors then crystallized the wild-type and mutant MECR proteins, and get structures similar to those of previously resolved members of this protein family. Comparison of wild-type and G165Q mutant structures shows differences in hydrogen bonding as well differences to the predicted modelled structure. Residues around the active center of MECR G165Q display much higher B factors than in wild-type suggesting higher mobility of these regions in variant. Since this made interpretations a bit weak, the authors then perform theoretical docking studies of different C chain length substrates to the crystallized proteins. In my view this approach does not convincingly replace the in vivo analysis of the mutant enzymes' capability to synthesize various carbon chain lengths. Finally, the authors try to find a molecular explanation for the proposed inability of mutated MECR to synthesize longer chain fatty acyl groups, i.e for the putative role of longer C chains in mitochondrial physiology, the key question of this paper. They show that the MECR G165Q mutant lacks detectable amounts of Cox1. This is indeed an interesting finding, yet raises the question whether this is due to a transcriptional effect (as seen in Ref. 4). Moreover, reading ref. 4, this leaves me puzzled why in this earlier work Cox1 protein is only slightly diminished (particularly after de novo protein synthesis) while in their new work the authors see a complete absence of Cox1 in the deletion and mutant strains. What about other

mitochondrially encoded proteins? What about protein synthesis activity in these cells (as done in Ref. 4). Overall, this is an interesting study, yet the lack of convincing data that longer carbon chain fatty acids are absent in the MECR mutant and why longer carbon chains may be needed to sustain respiratory growth of the mutant yeast cells still limits the impact of the study.

Response: We agree with the Reviewer that the analysis of the chain lengths synthesized by the mutant enzyme *in vivo* would be very informative and this question has puzzled us a lot. We have put a great effort to do the analysis. However, the $\Delta etr1$ yeast strains having ACP with a tag (like Strep-tag II in Angerer et al.) and carrying a plasmid expressing MECR G165Q variant is not viable in our hands so far. Thus, lack of proper yeast strains has prevented to do this analysis. Instead, we measured the PDH and KGDH activities in yeast mitochondrial extracts and noticed that G165Q mutation in MECR does not affect on these activities indicating that lipoic acid is present. This means that octanoyl-ACP was available.

Specific comments

1. Introduction: Please define MDR (to avoid confusion with multidrug resistance).

Response: MDR is defined now in Introduction to be medium-chain dehydrogenase / reductase.

2. Results start with a kind of summary what will be done. Is this really needed?

Response: The first paragraph of Results is removed and replace with two short introductory sentences.

3. Fig. 3: Lipoylation of Lat1 in part b is hardly detectable. Why? Please provide longer exposures. In a the panels seem to have been assembled from separate plates. This is not indicated properly.

Response: Lat1 and Kgd2 are not always equally lipoylated when analyzed with western blot using anti-lipoic acid antibody. Although the band from lipoylated Kgd2 is more prominent than lipoylated Lat1, lipoylated Lat1 is also visible when exposing for longer time as seen in new figure 3c. Both enzyme complexes are active (figures 5a and b) indicating that both Lat1 and Kgd2 are properly lipoylated. The panels in Figure 3a are assembled from two individual plates because all the samples did not fit on one plate. All plates contained the necessary controls (wild-type, $\Delta etr1$ and $\Delta etr1$ with plasmid expressing wild-type MECR) and were poured from the same medium. This is now indicated in Material and Methods: Yeast respiratory growth assay / Spotting assay-section.

4. Are the MECR G165Q cells rho zero or rho plus? If the first is true, this would explain why Cox1 is undetectable. A mitochondrial protein biosynthesis assay would be excellent to know what the effects in the various cells are.

Response: We tested if MECR G165Q cells are rho zero or rho plus by introducing back *Etr1* on a plasmid and tested the respiratory growth on non-fermentable medium (glycerol) (Supplementary figure 11). These cells were able to grow on the plates indicating that cells

must harbor an intact mitochondrial genome. Thus, loss of mitochondrial DNA does not explain why Cox1 is undetectable on our blots. It has been shown that mitochondrial ACP is associated with mitochondria in humans and trypanosomes and it is very likely this happens also in yeast. Our aim is to shed a light to this topic in future by using MECR G165Q mutant as a tool.

5. Fig. 4 (in my view) is more a supplement figure.

Response: Figure 4 is now Supplementary figure 4.

6. Fig. 5 and 6 can easily be combined because they address the same issue.

Response: Figures 5 and 6 are combined to be figure 4.

7. The paragraph on heme in Discussion is not really clear to me. Either this topic needs better explanation (what is the link of KGDH and heme) or - even better – heme content is measured (cytochromes, catalase).

Response: Heme synthesis requires succinyl-CoA that is generated by α -ketoglutarate dehydrogenase and catalase is a heme-containing enzyme. We measured catalase activities from wild-type, $\Delta etr1$ or $\Delta etr1$ expressing wild-type MECR or G165Q variant yeast spheroplasts (Figure 5d). There were no changes in catalase activities between these yeast strains indicating that heme content is not affected in MECR G165Q cells and thus lack of heme cannot explain the defect in respiration. In addition, we also show that α -ketoglutarate dehydrogenase is active in this strain and thus succinyl-CoA required to heme synthesis is produced (Figure 5b). The paragraph on heme in Discussion is completely revised (page 12).

Reviewer #3 (Remarks to the Author):

The manuscript entitled “An engineered variant of MECR reductase reveals indispensability of long-chain acyl-ACPs for mitochondrial respiration,” by Rahman, et al., offers an exploration into the mitochondrial FAS enoylreductase MECR. At its core, the manuscript attempts to better understand how MECR contributes to fatty acid biosynthesis and the metabolic requirements for both lipoic acid precursors and long chain fatty acids. While it has long been dogma that the mitochondrial FAS only produces octanoic acid to supply precursor for lipoylation, recent x-ray and cryoEM structures of other metabolic components, including the ribosome and iron-sulfur biosynthesis among others, indicate that long-chain fatty acids attached to the ACP participate in metabolic regulation. This manuscript sets out to clarify this fundamental contradiction.

The team used heterologous human MECR in a *Saccharomyces cerevisiae* $\Delta etr1$ yeast strain lacking native mtFAS enoyl reductase. Using in silico design of a mutant MECR suitable for octanoate biosynthesis, but unable to reduce longer chain fatty acids, the transgenic strain is used to evaluate the cellular effect of lacking long-chain fatty acids in the presence of endogenous lipoic acid, and thus the effect upon mitochondrial respiration with the presence of lipoylation. The authors claim to combine in silico, in vitro, and in vivo work to compare the stability and function of wild-type MECR and mutant MECR proteins.

However, the manuscript primarily details the computational studies in development and x-

ray crystal structure data collected with the mutant MECR. The in vivo portions of the work are limited to evaluation of mutants expressed in Δetr1 strain, along with anti-lipoic acid western blots and evaluation of growth on selected media. While these studies do indicate that the mutants are capable of producing octanoate for lipoic acid, the conclusion that they cannot form longer chain fatty acids is not fully established. More experiments are required to reach this conclusion. Overall, this manuscript addresses a central topic in our understanding of mtFAS that has major implications in mitochondrial metabolism and disease.

In addition to the above, the following items were identified that should be addressed:

1. The sentence “The structurally well characterized...” (Lines 35-37) lacks some transfer from last sentence introducing mtFAS.

Response: The sentence is corrected to be “The structurally well-characterized component of mtFAS, human 2*E*-enoyl-ACP reductase (MECR) rescues respiratory growth and lipoylation defects of a *Saccharomyces cerevisiae* Δetr1 strain lacking native mtFAS enoyl reductase.”

2. In Line 59, the word “adjust” seems not quite appropriate- might be better to be changed to “regulate”.

Response: The word “regulate” is now used instead of “adjust”.

3. There’s an extra dot in the annotation “(Figure 1a.)” (Line 61).

Response: Extra dot removed.

4. In the sentence “We have previously...” (Lines 118-121), the previous studies on three different mutations is quoted, but only two of them was mentioned in the subsequent discussion. Is there any specific reason F324Y is not included in this study?

Response: The mutation sites considered were based on analyzing the results of the previous mutational studies (e.g., Chen et al., *J. Mol. Biol.* (2008) 379, 830–844), but also on the analysis of the modelled MECR complex with the 2*E*-C16 substrate fragment generated in this work (Figure R1).

Figure R1. The location of the Phe324, Ile129 and Gly165 residues in the WT MECR structure with the docked 2*E*-C16 substrate fragment. The modelled substrate is shown in ball-and-stick representation, the tail carbons over C8 are in orange. A selected distance between the modelled substrate and Phe324 is shown in magenta (in Å).

The G165 and I129 mutation points were selected as the most promising ones, hence we focused the discussion on these. The F324 position, about which the Reviewer is asking, was not considered as promising enough in terms of blocking the substrate tunnel for the substrates longer than 2E-C8, which was the main aim of this paper. In the paper by Chen *et al.* the changes in kinetic parameters upon all the tested mutations were not significant enough, and they were not selective enough for the 2E-C8 substrates (e.g., for F324Y: 2E-C8 $K_m=5.5$ μM and $k_{\text{cat}}=3.3$ s^{-1} , 2E-C14 $K_m=8.3$ μM and $k_{\text{cat}}=2.1$ s^{-1}). From the visual inspection, we decided that the residue F324 is located too far away from the predicted substrate tail position to discriminate between 2E-C8 and longer substrates (Fig. R1) and it was already a bulky residue, so it could be hard to exchange it with a larger residue to block the substrate tunnel (Trp or Arg could be considered, but both show steric clashes with the other neighboring protein residues in the most frequent rotamers). The text on page 5 is modified correspondingly.

5. After the sentence “The backbone geometry...” (Lines 123-124), you may also consider mentioning the feasibility to work on I129.

Response: The backbone geometry of Gly165 was mentioned since Gly is more flexible due to the lack of the side chain, and less standard backbone geometries would therefore be allowed compared the other amino acids, to which it could be mutated. If, in the original structure, the backbone dihedrals for Gly were in the disfavored region for the mutated amino acid, then this would enforce the backbone conformational change that could destabilize the enzyme structure. In the case of Ile, there is no such problem, the Phi/Psi dihedral angles (-122.38°, 158.99°) are located in the center of the β -sheet region, since Ile129 is located at the end of the β -strand. We modified the text on page 5 by adding “, as would the backbone geometry at position 129 (-122 °, 159°)”.

6. A space missing in “glycerol,medium (SCG)” (Line 142).

Response: A space is added.

7. After the sentence “The K_m values...” (Lines 173-175), it would be better to have one sentence conclude the difference between the wild-type (decreases to 1/6) and mutant MECR (decreases to 1/14). Also, an enzyme with a lower K_m has a higher affinity for its substrate. It's not clear here why both types of MECR have a higher affinity towards C10 than C6, which may need to be explained explicitly here.

Response: The sentence “Thus, the K_m value of wild-type MECR with C10 substrate is 18% of the K_m value with C6 substrate, while in G165Q mutant enzyme the K_m value with C10 substrate is 7% of the K_m value with C6 substrate.” is added on page 6. The catalytic cycle is complicated and therefore understanding k_{cat} and K_m in the context of the structure is difficult. For example, it is not known if k_{cat} refers to a chemical step or to a conformational switch and/or to the off-dissociation of the products. Similarly, K_m depends in a complicated way on individual rate constants. Understanding at the molecular level of these K_m values is

therefore not straight forward. We have now added a sentence that the structural dynamical properties of wild type and G165Q variant are different, which is also a relevant for the catalytic properties (page 6).

8. Sentences from Line 223-231 and Lines 390-402 analyze the possible effect of hydrogen-bonding shift observed from crystallography. This would be more convincing if some relevant computational work be done to support authors' hypotheses.

Response: The crystal structure defines very well the side chain structure of Q165. The changes of the hydrogen bond interactions related to the interaction of the Q165 side chain with the P130 main chain oxygen are now better described (page 7). This appears to cause changes in the conformational flexibility properties in nearby residues, as discussed in the text of the manuscript. Such changes can be expected and they will effect the catalytic properties but is difficult to predict such changes in a quantitative way by further modelling calculations.

9. In Line 233, the term “apo” is used for the first time without explanation.

Response: The term “apo” was mistakenly used here instead of “wild-type” and this is now corrected to the text.

10. The paragraph “In previous studies...” (Line 404-410) seems not very logical. Admittedly, the lack of longer-chain fatty acids is probably the direct cause of the respiratory-defect phenotype, but we can't rule out the possibility that those yeast strains also cannot produce enough C8 precursor which is the prerequisite for longer-chain fatty acids. Also in Line 406, the term “ Δ mtFAS” should be “ Δ etr1”.

Response: To rule out the possibility that MECR G165Q mutant strain cannot produce enough octanoyl-ACP (C8), we measured the activities of lipoylation-dependent enzyme complexes pyruvate dehydrogenase and α -ketoglutarate dehydrogenase. These activities were on the same level in mitochondrial extracts from wild-type yeast and Δ etr1 expressing wild-type MECR or G165Q variant (Figures 5 a and b). Additionally, we ruled out the possibility that respiratory defects could be due to the heme-deficiency by analyzing the catalase activities in our yeast strains. Heme synthesis requires succinyl-CoA that is generated by α -ketoglutarate dehydrogenase and catalase is a heme-containing enzyme. There were no changes in catalase activities in wild-type, Δ etr1 or Δ etr1 expressing wild-type MECR or G165Q variant (Figure 5d). Because we can't detect any defect in synthesis of octanoyl-ACP that could lead to the respiratory deficient phenotype, the only possible explanation is lack of mitochondrial longer-chain fatty acids.

11. Lines 416-417, “due to a shortened binding cavity recovered cellular lipoic acid levels of Δ etr1 yeast cells.” This was not experimentally determined. Although lipoic acid was detected, the inability to prepare longer chains was not reported. More experiments are needed to evaluate exactly what fatty acids are made, and which are not. At a minimum, a global FAME analysis of the fatty acids in each strain in various culture conditions would prove valuable.

Response: As already discussed with Reviewer #2, we have tried to analyze directly mitochondrially produced acyl-ACPs, but lack of proper yeast strain has prevented us to do so. We analyzed total fatty acid profiles from each yeast strain (Supplementary figure 10) and we did not see any changes in these profiles between the strains. These results were expected because the role of mtFAS in generation of total cellular fatty acid pool is minor compared to cytosolic FAS I pathway. Discussion about points 10. and 11. is added on page 12.

12. The legend “MECR accepts...” (Lines 867-868) contains key information that should be provided in the main text. In the previous discussion, the object changed from acyl-ACP to acyl-CoA without any transfer, thus being confusing.

Response: This is very true, and now a sentence “*In vitro*, MECR accepts fatty acyl groups that are attached to either CoA or ACP via a thioester bond as substrates.” is added to the introduction (page 3).

13. In Line 896, the format of “(PDB-ID 7AYC)” is not consistent to other “PDB entry”.

Response: This text is changed to be (PDB entry 7AYC).

We hope that our responses to the reviewers’ comments are satisfactory and the manuscript can be accepted for publication in Nature Communications in this present revised form.

Sincerely Yours,

Kaija Autio

REVIEWER COMMENTS

Reviewer #1 (Remarks to the Author):

The authors have satisfactorily addressed all my concerns.

Therefore, I believe this excellent manuscript is ready for publication.

Reviewer #2 (Remarks to the Author):

Reviewer #2 (Remarks to the Author): See my "COMMENTS" for rebuttal text below.

.... Overall, this is an interesting study, yet the lack of convincing data that longer carbon chain fatty acids are absent in the MECR mutant and why longer carbon chains may be needed to sustain respiratory growth of the mutant yeast cells still limits the impact of the study.

Response: We agree with the Reviewer that the analysis of the chain lengths synthesized by the mutant enzyme in vivo would be very informative and this question has puzzled us a lot. We have put a great effort to do the analysis. However, the Δ etr1 yeast strains having ACP with a tag (like Strep-tag II in Angerer et al.) and carrying a plasmid expressing MECR G165Q variant is not viable in our hands so far. Thus, lack of proper yeast strains has prevented to do this analysis.

COMMENT: This is unfortunate, but I accept this unfortunate limitation.

Instead, we measured the PDH and KGDH activities in yeast mitochondrial extracts and noticed that G165Q mutation in MECR does not affect on these activities indicating that lipoic acid is present. This means that octanoyl-ACP was available.

COMMENT: The activities of PDH and KGDH was measured using fumarase and catalase as controls. The authors claim that both PDH and KGDH "Enzyme activities were on comparable level in the MECR G165Q mutant 310 strain, wild-type yeast mitochondria or the Δ etr1 strain expressing MECR, while complete loss of 311 mtFAS function(Δ etr1) caused a severe reduction in both activities." There is indeed a trend to justify this statement. However, the data quality is really poor (meaning NO publication quality), as all four enzyme activities bounce around from almost zero to a maximum for the three data sets. Something is wrong with these enzyme analyses, because from own experience I know that all four activities can be measured with +/- 20% differences, at most. I urge to get better data for this approach.

Specific comments

1. Introduction: Please define MDR (to avoid confusion with multidrug resistance).

Response: MDR is defined now in Introduction to be medium-chain dehydrogenase / reductase.

COMMENT: Resolved

2. Results start with a kind of summary what will be done. Is this really needed?

Response: The first paragraph of Results is removed and replaced with two short introductory sentences.

COMMENT: Resolved

3. Fig. 3: Lipoylation of Lat1 in part b is hardly detectable. Why? Please provide longer exposures. In the panels seem to have been assembled from separate plates. This is not indicated properly.

Response: Lat1 and Kgd2 are not always equally lipoylated when analyzed with western blot using anti-lipoic acid antibody. Although the band from lipoylated Kgd2 is more prominent than lipoylated Lat1, lipoylated Lat1 is also visible when exposing for longer time as seen in new figure 3c. Both enzyme complexes are active (figures 5a and b) indicating that both Lat1 and Kgd2 are properly lipoylated.

COMMENT: In my experience, this is not really true and lipoylation of Lat1 can be better resolved experimentally. The longer exposure helped only a bit to show the behavior of Lat1 in all these mutant cells.

The panels in Figure 3a are assembled from two individual plates because all the samples did not fit on one plate. All plates contained the necessary controls (wild-type, Δ etr1 and Δ etr1 with plasmid expressing wild-type MECR) and were poured from the same medium. This is now indicated in Material and Methods: Yeast respiratory growth assay / Spotting assay-section.

COMMENT: I accept this explanation, but this is NOT what I meant (sorry for not being clear). What I meant is that the use of different plates should be indicated by separating these parts by black or white lines (to really make clear that the results come from different, yet formally identical plates). ((I mention this, because some people search for such issues.))

4. Are the MECR G165Q cells rho zero or rho plus? If the first is true, this would explain why Cox1 is undetectable. A mitochondrial protein biosynthesis assay would be excellent to know what the effects in the various cells are.

Response: We tested if MECR G165Q cells are rho zero or rho plus by introducing back Etr1 on a plasmid and tested the respiratory growth on non-fermentable medium (glycerol) (Supplementary figure 11). These cells were able to grow on the plates indicating that cells must harbor an intact mitochondrial genome. Thus, loss of mitochondrial DNA does not explain why Cox1 is undetectable on our blots. It has been shown that mitochondrial ACP is associated with mitoribosomes in humans and trypanosomes and it is very likely this happens also in yeast. Our aim is to shed a light to this topic in future by using MECR G165Q mutant as a tool.

COMMENT: If the text of the authors is correct, the issue would be resolved. However, I have difficulties with the labeling of the sectors in Suppl Fig. 11. First, the two plates are not showing the strains in the same order (which is really confusing). Second, the labeling of the sectors mixes strain names with plasmids transformations. I urge the authors to be more didactic here.

5. Fig. 4 (in my view) is more a supplement figure.

Response: Figure 4 is now Supplementary figure 4.

COMMENT: Resolved

6. Fig. 5 and 6 can easily be combined because they address the same issue.

Response: Figures 5 and 6 are combined to be figure 4.

COMMENT: Resolved

7. The paragraph on heme in Discussion is not really clear to me. Either this topic needs better explanation (what is the link of KGDH and heme) or - even better – heme content is measured (cytochromes, catalase).

Response: Heme synthesis requires succinyl-CoA that is generated by α -ketoglutarate dehydrogenase and catalase is a heme-containing enzyme. We measured catalase activities from wild-type, Δ etr1 or Δ etr1 expressing wild-type MECR or G165Q variant yeast spheroplasts (Figure 5d). There were no changes in catalase activities between these yeast strains indicating that heme content is not affected in MECR G165Q cells and thus lack of heme cannot explain the defect in respiration. In addition, we also show that α -ketoglutarate dehydrogenase is active in this strain and thus succinyl-CoA required to heme synthesis is produced (Figure 5b). The paragraph on heme in Discussion is completely revised (page 12).

COMMENT: I cannot but repeat my concern about the data quality of Fig. 5d. Catalase is one of the easiest enzymes to measure. Why do the authors present such high fluctuations in the three data sets (up to factor of ten)? This is NOT publication quality. The experiments need to be repeated after optimizing the enzyme detection system.

Reviewer #3 (Remarks to the Author):

The authors have sufficiently addressed the reviewer concerns.

REVIEWER COMMENTS:

Below we give point-by-point responses for the remaining open questions, suggestions and concerns raised by reviewer #2 during the previous reviewing round.

Open question 1:

.... Overall, this is an interesting study, yet the lack of convincing data that longer carbon chain fatty acids are absent in the MECR mutant and why longer carbon chains may be needed to sustain respiratory growth of the mutant yeast cells still limits the impact of the study.

Response: We agree with the Reviewer that the analysis of the chain lengths synthesized by the mutant enzyme in vivo would be very informative and this question has puzzled us a lot. We have put a great effort to do the analysis. However, the Δ etr1 yeast strains having ACP with a tag (like Strep-tag II in Angerer et al.) and carrying a plasmid expressing MECR G165Q variant is not viable in our hands so far. Thus, lack of proper yeast strains has prevented to do this analysis.

COMMENT: This is unfortunate, but I accept this unfortunate limitation.

Instead, we measured the PDH and KGDH activities in yeast mitochondrial extracts and noticed that G165Q mutation in MECR does not affect on these activities indicating that lipoic acid is present. This means that octanoyl-ACP was available.

COMMENT: The activities of PDH and KGDH was measured using fumarase and catalase as controls. The authors claim that both PDH and KGDH “Enzyme activities were on comparable level in the MECR G165Q mutant 310 strain, wild-type yeast mitochondria or the Δ etr1 strain expressing MECR, while complete loss of 311 mtFAS function(Δ etr1) caused a severe reduction in both activities.” There is indeed a trend to justify this statement.

However, the data quality is really poor (meaning NO publication quality), as all four enzyme activities bounce around from almost zero to a maximum for the three data sets. Something is wrong with these enzyme analyses, because from own experience I know that all four activities can be measured with +/- 20% differences, at most. I urge to get better data for this approach.

Response to question 1: We repeated the PDH, α -KDH and fumarase assays by using four to six biological replicates and each assay was done duplicate. Figures 5a, 5b and 5c are presenting the received data.

Open question 2:

3. Fig. 3: Lipoylation of Lat1 in part b is hardly detectable. Why? Please provide longer exposures. In a the panels seem to have been assembled from separate plates. This is not indicated properly.

Response: Lat1 and Kgd2 are not always equally lipoylated when analyzed with western blot using anti-lipoic acid antibody. Although the band from lipoylated Kgd2 is more prominent than lipoylated Lat1, lipoylated Lat1 is also visible when exposing for longer time as seen in new figure 3c. Both enzyme complexes are active (figures 5a and b) indicating that both Lat1 and Kgd2 are properly lipoylated.

COMMENT: In my experience, this is not really true and lipoylation of Lat1 can be better

resolved experimentally. The longer exposure helped only a bit to show the behavior of Lat1 in all these mutant cells.

Response to question 2: We used now 12% resolving gel in SDS-PAGE for western blotting and run the gel long enough to really separate the bands of lipoylated Lat1 and Kgd2. Now both the bands are clearly visible and this is presented in a now figure 3b. The text for materials and methods, results and the figure legend are revised accordingly.

Open question 3:

The panels in Figure 3a are assembled from two individual plates because all the samples did not fit on one plate. All plates contained the necessary controls (wild-type, Δ etr1 and Δ etr1 with plasmid expressing wild-type MECR) and were poured from the same medium. This is now indicated in Material and Methods: Yeast respiratory growth assay / Spotting assay-section.

COMMENT: I accept this explanation, but this is NOT what I meant (sorry for not being clear). What I meant is that the use of different plates should be indicated by separating these parts by black or white lines (to really make clear that the results come from different, yet formally identical plates). ((I mention this, because some people search for such issues.))

Response to question 3: We are sorry for this misunderstanding. The black lines in figure 3a are now showing that we have used two identical plates to receive the result.

Open question 4:

4. Are the MECR G165Q cells rho zero or rho plus? If the first is true, this would explain why Cox1 is undetectable. A mitochondrial protein biosynthesis assay would be excellent to know what the effects in the various cells are.

Response: We tested if MECR G165Q cells are rho zero or rho plus by introducing back Etr1 on a plasmid and tested the respiratory growth on non-fermentable medium (glycerol) (Supplementary figure 11). These cells were able to grow on the plates indicating that cells must harbor an intact mitochondrial genome. Thus, loss of mitochondrial DNA does not explain why Cox1 is undetectable on our blots. It has been shown that mitochondrial ACP is associated with mitoribosomes in humans and trypanosomes and it is very likely this happens also in yeast. Our aim is to shed a light to this topic in future by using MECR G165Q mutant as a tool.

COMMENT: If the text of the authors is correct, the issue would be resolved. However, I have difficulties with the labeling of the sectors in Suppl Fig. 11. First, the two plates are not showing the strains in the same order (which is really confusing). Second, the labeling of the sectors mixes strain names with plasmids transformations. I urge the authors to be more didactic here.

Response to question 4: Supplementary figure 11. and the figure legend are now revised. The plates are showing the strains in the same order and the labeling the sectors is now corrected.

Open question 5:

7. The paragraph on heme in Discussion is not really clear to me. Either this topic needs better explanation (what is the link of KGDH and heme) or - even better – heme content is measured (cytochromes, catalase).

Response: Heme synthesis requires succinyl-CoA that is generated by α -ketoglutarate dehydrogenase and catalase is a heme-containing enzyme. We measured catalase activities from wild-type, Δ etr1 or Δ etr1 expressing wild-type MECR or G165Q variant yeast spheroplasts (Figure 5d). There were no changes in catalase activities between these yeast strains indicating that heme content is not affected in MECR G165Q cells and thus lack of heme cannot explain the defect in respiration. In addition, we also show that α -ketoglutarate dehydrogenase is active in this strain and thus succinyl-CoA required to heme synthesis is produced (Figure 5b). The paragraph on heme in Discussion is completely revised (page 12).

COMMENT: I cannot but repeat my concern about the data quality of Fig. 5d. Catalase is one of the easiest enzymes to measure. Why do the authors present such high fluctuations in the three data sets (up to factor of ten)? This is NOT publication quality. The experiments need to be repeated after optimizing the enzyme detection system.

Response to question 5: For catalase activity assays we followed now the method described by Chance and Maehly in Methods in Enzymology (vol. 2, page 764-775). For measurements we used five to six biological repeats that were measured duplicate each. The data is now presented in figure 5d and the text in materials and methods is revised accordingly.

We hope that these responses to the reviewer's comments are satisfactory and the manuscript can be accepted for publication in Nature Communications in this present revised form.

Sincerely Yours,

Kaija Autio

REVIEWERS' COMMENTS

Reviewer #2 (Remarks to the Author):

Review of re-revised manuscript (see ANSWER)

Open question 1:

COMMENT: The activities of PDH and KGDH was measured using fumarase and catalase as controls. The authors claim that both PDH and KGDH “Enzyme activities were on comparable level in the MECR G165Q mutant 310 strain, wild-type yeast mitochondria or the Δ etr1 strain expressing MECR, while complete loss of 311 mtFAS function(Δ etr1) caused a severe reduction in both activities.” There is indeed a trend to justify this statement. However, the data quality is really poor (meaning NO publication quality), as all four enzyme activities bounce around from almost zero to a maximum for the three data sets. Something is wrong with these enzyme analyses, because from own experience I know that all four activities can be measured with +/- 20% differences, at most. I urge to get better data for this approach.

Response to question 1: We repeated the PDH, α -KDH and fumarase assays by using four to six biological replicates and each assay was done duplicate. Figures 5a, 5b and 5c are presenting the received data.

ANSWER: This looks very nice now.

Open question 2:

COMMENT: In my experience, this is not really true and lipoylation of Lat1 can be better resolved experimentally. The longer exposure helped only a bit to show the behavior of Lat1 in all these mutant cells.

Response to question 2: We used now 12% resolving gel in SDS-PAGE for western blotting and run the gel long enough to really separate the bands of lipoylated Lat1 and Kgd2. Now both the bands are clearly visible and this is presented in a now figure 3b. The text for materials and methods, results and the figure legend are revised accordingly.

ANSWER: The resolution of Lat1-LA looks better now. However, the authors write “The results show that mutations in G165 do not affect protein lipoylation, indicating that MECR mutants G165H, G165L, G165Q and G165F are able to synthesize fatty acids at least up to C8.” This maybe more or less true for lipoylation of Kgd1, but the now visible data shows that this statement is certainly NOT true for Lat1. How is this disparity explained, and why is this important point not addressed? Minor point: There is a typo in Fig. 3b: Kgd1 instead of kgd1.

Open question 3:

COMMENT: I accept this explanation, but this is NOT what I meant (sorry for not being clear). What I meant is that the use of different plates should be indicated by separating these parts by black or white lines (to really make clear that the results come from different, yet formally identical plates). ((I mention this, because some people search for such issues.))

Response to question 3: We are sorry for this misunderstanding. The black lines in figure 3a are now showing that we have used two identical plates to receive the result.

ANSWER: Resolved.

Open question 4:

COMMENT: If the text of the authors is correct, the issue would be resolved. However, I have difficulties with the labeling of the sectors in Suppl Fig. 11. First, the two plates are not showing the strains in the same order (which is really confusing). Second, the labeling of the sectors mixes strain names with plasmids transformations. I urge the authors to be more didactic here.

Response to question 4: Supplementary figure 11. and the figure legend are now revised. The plates are showing the strains in the same order and the labeling the sectors is now corrected.

ANSWER: Resolved.

Open question 5:

COMMENT: I cannot but repeat my concern about the data quality of Fig. 5d. Catalase is one of the easiest enzymes to measure. Why do the authors present such high fluctuations in the three data sets (up to factor of ten)? This is NOT publication quality. The experiments need to be repeated after optimizing the enzyme detection system.

Response to question 5: For catalase activity assays we followed now the method described by Chance and Maehly in *Methods in Enzymology* (vol. 2, page 764-775). For measurements we used five to six biological repeats that were measured duplicate each. The data is now presented in figure 5d and the text in materials and methods is revised accordingly.

ANSWER: Resolved (as commented above for Fig. 5a-c).

REVIEWERS' COMMENTS

Reviewer #2 (Remarks to the Author):

The resolution of Lat1-LA looks better now. However, the authors write “The results show that mutations in G165 do not affect protein lipoylation, indicating that MECR mutants G165H, G165L, G165Q and G165F are able to synthesize fatty acids at least up to C8.” This maybe more or less true for lipoylation of Kgd1, but the now visible data shows that this statement is certainly NOT true for Lat1. How is this disparity explained, and why is this important point not addressed? Minor point: There is a typo in Fig. 3b: Kgd1 instead of kgd1.

Response: We thank the reviewer for this critical comment. We have now revised the original strong statement and the current revised sentence reads now “The results show that mutations in G165 allow protein lipoylation, indicating that MECR mutants G165H, G165L, G165Q and G165F are able to synthesize fatty acids at least up to C8.” Typo in fig. 3b is corrected.

Sincerely Yours,

Kaija Autio